# Dense Associative Memory Through the Lens of Random Features

**Benjamin Hoover**
IBM Research & Georgia Tech
benjamin.hoover@ibm.com

**Duen Horng Chau**
Georgia Tech
polo@gatech.edu

**Hendrik Strobelt**
IBM Research & MIT-IBM
hendrik.strobelt@ibm.com

**Parikshit Ram**
IBM Research
parikshit.ram@ibm.com

**Dmitry Krotov**
IBM Research
krotov@ibm.com

## Abstract

Dense Associative Memories are high storage capacity variants of the Hopfield networks that are capable of storing a large number of memory patterns in the weights of the network of a given size. Their common formulations typically require storing each pattern in a separate set of synaptic weights, which leads to the increase of the number of synaptic weights when new patterns are introduced. In this work we propose an alternative formulation of this class of models using random features, commonly used in kernel methods. In this formulation the number of network's parameters remains fixed. At the same time, new memories can be added to the network by modifying existing weights. We show that this novel network closely approximates the energy function and dynamics of conventional Dense Associative Memories and shares their desirable computational properties.

## 1   Introduction

Hopfield network of associative memory is an elegant mathematical model that makes it possible to store a set of memory patterns in the synaptic weights of the neural network [1]. For a given prompt $\sigma_i(t = 0)$, which serves as the initial state of that network, the neural update equations drive the dynamical flow towards one of the stored memories. For a system of $K$ memory patterns in the $D$-dimensional binary space the network's dynamics can be described by the temporal trajectory $\sigma_i(t)$, which descends the energy function

$$E = -\sum_{\mu=1}^{K} \Big( \sum_{i=1}^{D} \xi_i^{\mu} \sigma_i \Big)^2 \tag{1}$$

Here $\xi_i^{\mu}$ (index $\mu = 1...K$, and index $i = 1...D$) represent memory vectors. The neural dynamical equations describe the energy descent on this landscape. In this formulation, which we call the **memory representation**, the geometry of the energy landscape is encoded in the weights of the network $\xi_i^{\mu}$, which coincide with the memorised patterns. Thus, in situations when the set of the memories needs to be expanded by introducing new patterns one must introduce additional weights.

Alternatively, one could rewrite the above energy in a different form, which is more commonly used in the literature. Specifically, the sum over the memories can be computed upfront and the energy can be written as

$$E = -\sum_{i,j=1}^{D} T_{ij} \sigma_i \sigma_j, \quad \text{where} \quad T_{ij} = \sum_{\mu=1}^{K} \xi_i^{\mu} \xi_j^{\mu} \tag{2}$$

38th Conference on Neural Information Processing Systems (NeurIPS 2024).

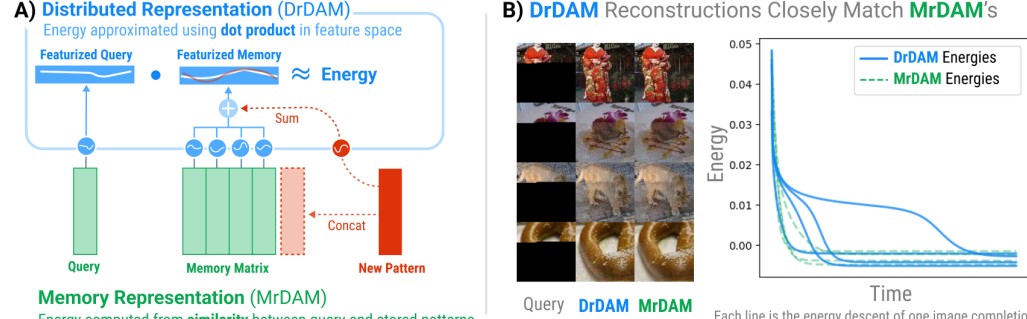

Figure 1: The **D**istributed **R**epresentation for **D**ense **A**ssociative **M**emory (**DrDAM**) approximates both the energy and fixed-point dynamics of the traditional **M**emory **R**epresentation for **D**ense **A**ssociative **M**emory (**MrDAM**) while having a parameter space of constant size. A) Diagram of DrDAM using a ⊙ **basis function** parameterized by random features (e.g., see eq. (8)). In the distributed representation, adding new memories does not change the size of the memory tensor. B) Comparing energy descent dynamics between DrDAM and MrDAM on 3x64x64 images from Tiny Imagenet [11]. Both models are initialized on queries where the bottom two-thirds of pixels are occluded with zeros; dynamics are run while clamping the visible pixels and their collective energy traces shown. DrDAM achieves the same fixed points as MrDAM, and these final fixed points have the same energy. The energy decreases with time for both MrDAM and DrDAM, although the dependence of the energy relaxation towards the fixed point is sometimes different between the two representations. Experimental setup is described in appendix D.

In this form one can think about weights of the network being the symmetric tensor $T_{ij}$ instead of $\xi_i^\mu$. One advantage of formulating the model this way is that the tensor $T_{ij}$ does not require adding additional parameters when new memories are introduced. Additional memories are stored in the already existing set of weights by redistributing the information about new memories across the already existing network parameters. We refer to this formulation as **distributed representation**.

A known problem of the network (eqs. (1) and (2)) is that it has a small memory storage capacity, which scales at best linearly as the size of the network $D$ is increased [1]. This limitation has been resolved with the introduction of Dense Associative Memories (DenseAMs), also known as Modern Hopfield Networks [2]. This is achieved by strengthening the non-linearities (interpreted as neural activation functions) in eq. (1), which can lead to the super-linear and even exponentially large memory storage capacity [2, 3]. Using continuous variables $\mathbf{x} \in \mathbb{R}^D$, the energy is defined as[1]

$$E = -Q\Big[ \sum_{\mu=1}^{K} F\Big( S\big[\boldsymbol{\xi}^\mu, \mathbf{g}(\mathbf{x})\big] \Big) \Big], \tag{3}$$

where the function $\mathbf{g} : \mathbb{R}^D \to \mathbb{R}^D$ is a vector function (e.g., a sigmoid, a linear function, or a layernorm), the function $F(\cdot)$ is a rapidly growing separation function (e.g., power $F(\cdot) = (\cdot)^n$ or exponent), $S[\mathbf{x}, \mathbf{x}']$ is a similarity function (e.g., a dot product or a Euclidean distance), and $Q$ is a scalar monotone function (e.g., linear or logarithm). For instance, in order to describe the classical Hopfield network with binary variables (eq. (1)) one could take: linear $Q$, quadratic $F(\cdot) = (\cdot)^2$, dot product $S$, and a sign function for $g_i = sign(x_i) = \sigma_i$. There are many possible combinations of various functions $\mathbf{g}, F(\cdot), S(\cdot, \cdot)$ that lead to different models from the DenseAM family [2–7]; many of the resulting models have proven useful for various problems in AI and neuroscience [8]. Diffusion models have been linked to even more sophisticated forms of the energy landscape [9, 10].

From the perspective of the information storage capacity DenseAMs are significantly superior compared to the classical Hopfield networks. At the same time, most[2] of the models from the DenseAM family are typically formulated using the memory representation, and for this reason

---

[1]Throughout the paper we use bold symbols for denoting vectors and tensors, e.g., $\boldsymbol{\xi}^\mu$ is a $D$-dimensional vector in the space of neurons for each value of index $\mu$. Individual elements of those vectors and tensors are denoted with the same symbol, but with plain font. In the example above, these individual elements have an explicit vector index, e.g., $\xi_i^\mu$. Same applies to vectors in the feature space introduced later.

[2]This is true for all DenseAMs with the exception of the power model of Krotov and Hopfield [2], which can be written using n-index tensors $T_{i_1, i_2, \ldots, i_n}$ in analogy with the 2-tensor $T_{ij}$ as in eq. (2).

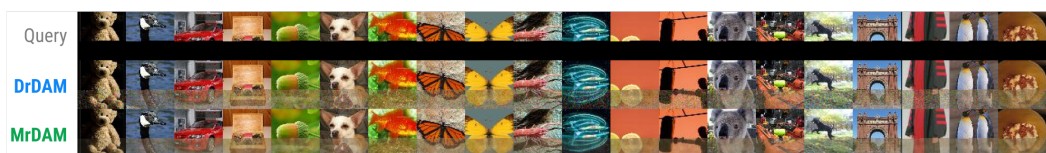

Figure 2: DrDAM achieves parameter compression over MrDAM, successfully storing 20 different 64x64x3 images from TinyImagenet [11] and retrieving them when occluding the lower 40% of each query. The memory matrix of MrDAM is of shape $(20, 12288)$ while the memory tensor of DrDAM is of shape $Y = 2 \cdot 10^5$, a $\sim 20\%$ reduction in the number of parameters compared to MrDAM; all other configurations for this experiment match those in appendix D. Further compression can be achieved with a higher tolerance for DrDAM's retrieval error, smaller $\beta$, and fewer occluded pixels, see § 4. **Top:** Occluded query images. **Middle:** Fixed-point retrievals from DrDAM. **Bottom:** (ground truth) Fixed-point retrievals of MrDAM.

require introduction of new weights when additional memory patterns are added to the network. The main question that we ask in our paper is: *how can we combine superior memory storage properties of DenseAMs with the distributed (across synaptic weights) formulation of these models in the spirit of classical Hopfield networks* (eq. (2))? If such a formulation is found, it would allow us to add memories to the existing network by simply recomputing already existing synaptic weights, without adding new parameters.

A possible answer to this question is offered by the theory of random features and kernel machines. Given an input domain $\mathcal{X}$, kernel machines leverage a positive definite *Mercer kernel function* $k : \mathcal{X} \times \mathcal{X} \to \mathbb{R}_+$ that measures the similarity between pairs of inputs. The renowned "kernel trick" allows one to compute the inner-product

$$k(\mathbf{x}, \mathbf{x}') = \langle \boldsymbol{\varphi}(\mathbf{x}), \boldsymbol{\varphi}(\mathbf{x}') \rangle = \sum_{\alpha=1}^{Y} \varphi_\alpha(\mathbf{x}) \varphi_\alpha(\mathbf{x}') \tag{4}$$

between two inputs $\mathbf{x}, \mathbf{x}' \in \mathcal{X}$ in a rich feature space defined by the feature map $\boldsymbol{\varphi}(\mathbf{x})$ without ever explicitly realizing the feature map $\boldsymbol{\varphi}(\mathbf{x})$. Various machine learning models (such as support vector machines [12], logistic regression, and various others [13, 14]) can be learned with just access to pairwise inner-products, and thus, the kernel trick allows one to learn such models in an extremely expressive feature space. Kernel functions have been developed for various input domains beyond the Euclidean space such as images, documents, strings (such as protein sequences [15]), graphs (molecules [16], brain neuron activation paths) and time series (music, financial data) [17]. Common kernels for Euclidean data are the radial basis function or RBF kernel $k(\mathbf{x}, \mathbf{x}') = \exp(-\gamma \|\mathbf{x} - \mathbf{x}'\|_2^2)$ and the polynomial kernel $k(\mathbf{x}, \mathbf{x}') = (\langle \mathbf{x}, \mathbf{x}' \rangle + b)^p$. To appreciate the expressivity of these kernel machines, note that, for input domain $\mathbb{R}^D$, the RBF kernel corresponds to an infinite dimensional feature space ($Y = \infty$) and the polynomial kernel to a $O(D^p)$ dimensional feature space.

Interpreting the composition of the separation and similarity functions in eq. (3) as the left hand side of the kernel trick eq. (4) we can map the energy into the feature space, using appropriately chosen feature maps. Subsequently, the order of the summations over memories and features can be swapped, and the sum over memories can be computed explicitly. This makes it possible to encode all the memories in a tensor $T_\alpha$, which we introduce in section § 3, that contains all the necessary information about the memories. The energy function then becomes defined in terms of this tensor only, as opposed to individual memories. This functionality is summarized in fig. 1. Additionally, we show examples of retrieved Tiny ImageNet images that have been memorised using the original DenseAM model, which we call MrDAM, and the "featurized" version of the same model, which we call DrDAM (please see the explanations of these names in the caption to fig. 1). These examples visually illustrate that mapping the problem into the feature space preserves most of the desirable computational properties of DenseAMs, which normally are defined in the "kernel space".

**Contributions:**

▶ We propose a novel approximation of a DenseAM network utilizing random features commonly used in kernel machines. This novel architecture does not require the storage of the original memories, and can incorporate new memories without increasing the size of the network.

▶ We precisely characterize the approximation introduced in the energy descent dynamics by this architecture, highlighting the different critical factors that drive the difference between the exact energy descent and the proposed approximate one.
▶ We validate our theoretical guarantee with empirical evaluations.

In the past, kernel trick has been used for optimizing complexity of the attention mechanism in Transformers [18], and those results have been recently applied to associative memory [19], given the various connections between Transformers and DenseAMs [4, 20]. Existing studies [18, 19] focus on settings when attention operation or associative memory retrieval is done in a single step update. This is different from our goals here, which is to study the recurrent dynamics of the associative memory updates and convergence of that dynamics to the attractor fixed points. Iatropoulos et al. [21] propose kernel memory networks which are a recurrent form of a kernel support vector machine, and highlight that DenseAM networks are special cases of these kernel memory networks. Making a connection between nonparametric kernel regression and associative memory, Hu et al. [22] propose a family of provably efficient sparse Hopfield networks [23, 24], where the dynamics of any given input are explicitly driven by a subset of the memories due to various entropic regularizations on the energy. DenseAMs have been also used for sequences [25, 26, 24]. To reduce the complexity of computing all the pairs of $F(S[\boldsymbol{\xi}, \mathbf{x}])$ for a given set of memories and queries, Hu et al. [27] leverage a low-rank approximation of this separation-similarity matrix using polynomial expansions. The kernel trick has also recently been used to increase separation between memories (with an additional learning stage to learn the kernel), thereby improving memory capacity [28]. There are also very recent theoretical analysis of the random feature Hopfield networks [29, 30], where their focus in on the construction of memories using random features. Kernels are also related to density estimation [31], and recent works have leveraged a connection between mixtures of Gaussians and DenseAMs for clustering [32, 33]. Lastly, random features have been used for biological implementations of both Transformers and DenseAMs [34, 35].

To the best of our knowledge there is no rigorous theoretical and empirical comparison of DenseAMs and their distributed (featurized) variants in recurrent memory storage and retrieval settings, as well as results pertaining to the recovery of the fixed points of the energy descent dynamics. This is the main focus of our work.

## 2   Technical background

Given the energy function in eq. (3), a variable $\mathbf{x}$ is updated in the forward pass through the "layers" of this recurrent model such that its energy decreases with each update. If the energy is bounded from below, this ensures that the input will (approximately) converge to a local minimum. This can be achieved by performing a "gradient descent" in the energy landscape. Considering the continuous dynamics, updating the input $\mathbf{x}$ over time with $d\mathbf{x}/dt$, we need to ensure that $dE/dt < 0$. This can be achieved by setting $d\mathbf{x}/dt \propto -\nabla_{\mathbf{x}} E$.

Discretizing the above dynamics, the update of an input $\mathbf{x}$ at the $t$-th recurrent layer is given by:

$$\mathbf{x}^{(t)} \leftarrow \mathbf{x}^{(t-1)} - \eta^{(t-1)} \nabla_{\mathbf{x}} E^{(t-1)}, \tag{5}$$

where $\eta^{(t)}$ is a (step dependent) step-size for the energy gradient descent, $E^{(t)}$ is the energy of the input after the $t$-th layer, and the input to the first layer $\mathbf{x}^{(0)} \leftarrow \mathbf{x}$. The final output of the associative memory network after $L$ layers is $\mathbf{x}^{(L)}$.

DenseAMs significantly improve the memory capacity of the associative memory network by utilizing rapidly growing nonlinearity-based separation-similarity compositions such as $F(S[\mathbf{x}, \boldsymbol{\xi}^{\mu}]) = \exp(\beta \langle \mathbf{x}, \boldsymbol{\xi}^{\mu} \rangle)$ or $F(S[\mathbf{x}, \boldsymbol{\xi}^{\mu}]) = \exp(-\beta/2 \|\mathbf{x} - \boldsymbol{\xi}^{\mu}\|_2)$ or $F(S[\mathbf{x}, \boldsymbol{\xi}^{\mu}]) = (\langle \mathbf{x}, \boldsymbol{\xi}^{\mu} \rangle)^p, p > 2$, among other choices, with $\beta > 0$ corresponding to the *inverse temperature* that controls how rapidly the separation-similarity function grows. However, these separation-similarity compositions do not allow for the straightforward simplifications as in eq. (2), except for the power composition. For a general similarity function, the update based on gradient descent over the energy in eq. (3) is given by:

$$\nabla_{\mathbf{x}} E = -\left. \frac{dQ(y)}{dy} \right|_{y=\sum_\mu F(S[\boldsymbol{\xi}^\mu, \mathbf{g}(\mathbf{x})])} \sum_{\mu=1}^{K} \left( \left. \frac{dF(s)}{ds} \right|_{s=S[\boldsymbol{\xi}^\mu, \mathbf{g}(\mathbf{x})]} \cdot \left. \frac{dS(\boldsymbol{\xi}^\mu, \mathbf{z})}{d\mathbf{z}} \right|_{\mathbf{z}=\mathbf{g}(\mathbf{x})} \cdot \frac{d\mathbf{g}(\mathbf{x})}{d\mathbf{x}} \right) \tag{6}$$

For example, with $Q(\cdot) = (1/\beta)\log(\cdot)$, $F(\cdot) = \exp(\beta\cdot)$, $S[\boldsymbol{\xi}^\mu, \mathbf{x}] = \langle\boldsymbol{\xi}^\mu, \mathbf{x}\rangle$ and $\mathbf{g}(\mathbf{x}) = \mathbf{x}/\|\mathbf{x}\|_2$, the energy function and the corresponding update[3] are:

$$E(\mathbf{x}) = -\frac{1}{\beta}\log\sum_{\mu=1}^{K}\exp(\beta\langle\boldsymbol{\xi}^\mu, \mathbf{g}(\mathbf{x})\rangle), \quad \nabla_\mathbf{x} E(\mathbf{x}) = -\frac{\sum_{\mu=1}^{K}\exp(\beta\langle\boldsymbol{\xi}^\mu, \mathbf{g}(\mathbf{x})\rangle)\boldsymbol{\xi}^\mu}{\sum_{\mu=1}^{K}\exp(\beta\langle\boldsymbol{\xi}^\mu, \mathbf{g}(\mathbf{x})\rangle)}\cdot\frac{d\mathbf{g}(\mathbf{x})}{d\mathbf{x}}. \quad (7)$$

This form does not directly admit itself to a distributed storage of memories as in eq. (2), and thus, in order to perform the gradient descent on the energy, it is necessary to keep all the memories in their original form. We will try to address this issue by taking inspiration from the area of *kernel machines* [36].

## 2.1 Random Features for Kernel Machines

The expressivity of kernel learning usually comes with increased computational complexity both during training and inference, taking time quadratic and linear in the size of the training set respectively. The groundbreaking work of Rahimi and Recht [37] introduced random features to generate explicit feature maps $\boldsymbol{\varphi}: \mathbb{R}^D \to \mathbb{R}^Y$ for the RBF and other *shift-invariant* kernels[4] that approximate the true kernel function – that is $\langle\boldsymbol{\varphi}(\mathbf{x}), \boldsymbol{\varphi}(\mathbf{x}')\rangle \approx k(\mathbf{x}, \mathbf{x}')$. Various such random maps have been developed for shift-invariant kernels [18, 19, 38] and polynomials kernels [39–41].

For the RBF kernel and the exponentiated dot-product or EDP kernel $k(\mathbf{x}, \mathbf{x}') = \exp(\langle\mathbf{x}, \mathbf{x}'\rangle)$, there are usually two classes of random features – trigonometric features and exponential features. For the RBF kernel $k(\mathbf{x}, \mathbf{x}') = \exp(-\|\mathbf{x} - \mathbf{x}'\|_2^2/2)$, the trigonometric features [37] are given on the left and the exponential features [18] are on the right:

$$\boldsymbol{\varphi}(\mathbf{x}) = \frac{1}{\sqrt{Y}}\begin{bmatrix}\cos(\langle\boldsymbol{\omega}^1, \mathbf{x}\rangle)\\\sin(\langle\boldsymbol{\omega}^1, \mathbf{x}\rangle)\\\ldots,\\\cos(\langle\boldsymbol{\omega}^Y, \mathbf{x}\rangle)\\\sin(\langle\boldsymbol{\omega}^Y, \mathbf{x}\rangle)\end{bmatrix}, \quad \boldsymbol{\varphi}(\mathbf{x}) = \frac{\exp(-\|\mathbf{x}\|_2^2)}{\sqrt{2Y}}\begin{bmatrix}\exp(+\langle\boldsymbol{\omega}^1, \mathbf{x}\rangle)\\\exp(-\langle\boldsymbol{\omega}^1, \mathbf{x}\rangle)\\\ldots,\\\exp(+\langle\boldsymbol{\omega}^Y, \mathbf{x}\rangle)\\\exp(-\langle\boldsymbol{\omega}^Y, \mathbf{x}\rangle)\end{bmatrix}, \quad (8)$$

where $\boldsymbol{\omega}^\alpha \sim \mathcal{N}(0, I_D)\forall\alpha \in \{1, \ldots, Y\}$ are the random projection vectors.[5] A random feature map $\boldsymbol{\varphi}$ for the RBF can be used for the EDP kernel by scaling $\boldsymbol{\varphi}(\mathbf{x})$ with $\exp(\|\mathbf{x}\|_2^2/2)$. While the trigonometric features ensure that $k(\mathbf{x}, \mathbf{x}) = \langle\boldsymbol{\varphi}(\mathbf{x}), \boldsymbol{\varphi}(\mathbf{x})\rangle = 1$, the exponential features ensure that $\boldsymbol{\varphi}(\mathbf{x}) \in \mathbb{R}_+^{2Y}$, which is essential in certain applications as in transformers [18, 19]. Furthermore, while the random samples $\boldsymbol{\omega}^\alpha \sim \mathcal{N}(0, I_D)$ are supposed to be independent, Choromanski et al. [42] show that the $\{\boldsymbol{\omega}^1, \ldots, \boldsymbol{\omega}^Y\}$ can be entangled to be exactly orthogonal to further reduce the variance of the approximation while maintaining unbiasedness. In general, the approximation of the random feature map is $O(\sqrt{D/Y})$, implying that a feature space with $Y \sim O(D/\epsilon^2)$ random features will ensure, with high probability, for any $\mathbf{x}, \mathbf{x}' \in \mathbb{R}^D$, $|k(\mathbf{x}, \mathbf{x}') - \langle\boldsymbol{\varphi}(\mathbf{x}), \boldsymbol{\varphi}(\mathbf{x}')\rangle| \leq \epsilon$. Scaling in the kernel functions such as $\exp(-\beta\|\mathbf{x} - \mathbf{x}'\|_2^2/2)$ or $\exp(\beta\langle\mathbf{x}, \mathbf{x}'\rangle)$ can be handled with the aforementioned random feature maps $\boldsymbol{\varphi}$ by applying them to $\sqrt{\beta}\mathbf{x}$ with $\langle\boldsymbol{\varphi}(\sqrt{\beta}\mathbf{x}), \boldsymbol{\varphi}(\sqrt{\beta}\mathbf{x}')\rangle \approx \exp(-\beta\|\mathbf{x} - \mathbf{x}'\|_2^2/2)$.

## 3 DrDAM with Random Features

Revisting the general energy function in eq. (3), if we have available an explicit mapping $\boldsymbol{\varphi}: \mathbb{R}^D \to \mathbb{R}^Y$ such that $\langle\boldsymbol{\varphi}(\boldsymbol{\xi}^\mu), \boldsymbol{\varphi}(\mathbf{x})\rangle \approx F(S[\boldsymbol{\xi}^\mu, \mathbf{x}])$, then we can simplify the general energy function in eq. (3) to

$$E(\mathbf{x}) \approx \hat{E}(\mathbf{x}) = -Q\left(\sum_{\mu=1}^{K}\langle\boldsymbol{\varphi}(\boldsymbol{\xi}^\mu), \boldsymbol{\varphi}(\mathbf{g}(\mathbf{x}))\rangle\right) = -Q\left(\left\langle\sum_{\mu=1}^{K}\boldsymbol{\varphi}(\boldsymbol{\xi}^\mu), \boldsymbol{\varphi}(\mathbf{g}(\mathbf{x}))\right\rangle\right). \quad (9)$$

---

[3]We are eliding the $d\mathbf{g}(\mathbf{x})/d\mathbf{x} = (1/\|\mathbf{x}\|_2)[I_D - (1/\|\mathbf{x}\|_2^{3/2})\mathbf{x}\mathbf{x}^\top]$ term for the ease of exposition.

[4]Kernel functions that only depend on $(\mathbf{x} - \mathbf{x}')$ and not individually on $\mathbf{x}$ and $\mathbf{x}'$.

[5]A technical detail here is that while we are using $Y$ random samples, we are actually developing a $2Y$-dimensional feature map $\boldsymbol{\varphi}: \mathbb{R}^D \to \mathbb{R}^{2Y}$ – we can get a $Y$ dimensional feature map by dropping the $\sin(\cdot)$ terms in the trigonometric features (and add a random rotation term $b^\alpha, \alpha \in [\![Y]\!]$ to the $\cos(\langle\boldsymbol{\omega}^\alpha, \mathbf{x}\rangle + b^\alpha)$ term), and the $\exp(-\cdot)$ term in the exponential features. This modification (using $2Y$ features instead of $Y$) reduces the variance of the kernel function approximation [18, Lemma 1, 2].

Denoting $\mathbf{T} = \sum_\mu \boldsymbol{\varphi}(\boldsymbol{\xi}^\mu)$, we can write a simplified general update step for any input $\mathbf{x}$ as:

$$\nabla_{\mathbf{x}}\hat{E} = -\left.\frac{dQ(s)}{ds}\right|_{s=\langle\boldsymbol{\varphi}(\mathbf{g}(\mathbf{x})),\mathbf{T}\rangle} \cdot \left(\left.\frac{\boldsymbol{\varphi}(\mathbf{z})}{d\mathbf{z}}\right|_{\mathbf{z}=\mathbf{g}(\mathbf{x})}^{\top}\mathbf{T}\right) \cdot \frac{d\mathbf{g}(\mathbf{x})}{d\mathbf{x}} \tag{10}$$

where $d\boldsymbol{\varphi}(\mathbf{x})/d\mathbf{x} \in \mathbb{R}^{Y\times D}$ is the gradient of the feature map with respect to its input. In the presence of such an explicit map $\boldsymbol{\varphi}$, we can distribute the memory in a MrDAM into the single $Y$-dimensional vector $\mathbf{T}$, and be able to apply the update in eq. (10). We can then use the random feature based energy gradient $\nabla_{\mathbf{x}}\hat{E}(\mathbf{x})$ instead of the true energy gradient $\nabla_{\mathbf{x}}E(\mathbf{x})$ in the energy gradient descent step in eq. (5).[6] We name this scheme "**D**istributed **r**epresentation for **D**ense **A**ssociative **M**emory" or DrDAM, and we compare the computational costs of DrDAM with the "**M**emory **r**epresentation of **D**ense **A**ssociative **M**emory" or MrDAM in the following:

**Proposition 1.** *With access to the $K$ memories $\{\boldsymbol{\xi}^\mu \in \mathbb{R}^D, \mu \in [\![K]\!]\}$, MrDAM takes $O(LKD)$ time and $O(KD)$ peak memory for $L$ energy gradient descent steps (or layers) as defined in eq. (5) with the true energy gradient $\nabla_{\mathbf{x}}E(\mathbf{x})$.*

Naively, the random feature based DrDAM would require $O(DY)$ memory to store the random vectors and the $\nabla_{\mathbf{x}}\boldsymbol{\varphi}(\mathbf{x})$ matrix. However, we can show that we can generate the random vectors on demand to reduce the overall peak memory to just $O(Y)$. The various procedures in DrDAM are detailed in Algorithm 1. The RF subroutine generates the random feature for any memory or input. The ProcMems subroutine consolidates all the memories into a single $\mathbf{T} \in \mathbb{R}^Y$ vector. The GradComp subroutine compute the gradient $\nabla_{\mathbf{x}}\hat{E}$. The following are the computational complexities of these procedures:

**Proposition 2.** *The RF subroutine in Algorithm 1 takes $O(DY)$ time and $O(D+Y)$ peak memory.*

**Proposition 3.** *ProcMems in Algorithm 1 takes $O(DYK)$ time and $O(D+Y)$ peak memory.*

**Proposition 4.** *GradComp in Algorithm 1 takes $O(D(Y+D))$ time and $O(D+Y)$ peak memory.*

Thus, the computational complexities of DrDAM neural dynamics are (see appendix F.1 for proof and discussions):

**Theorem 1.** *With a random feature map $\boldsymbol{\varphi}$ utilizing $Y$ random projections $\{\varphi_\alpha, \alpha \in \{1,\ldots,Y\}\}$ and $K$ memories $\{\boldsymbol{\xi}^\mu \in \mathbb{R}^D, \mu \in \{1,\ldots,K\}\}$, the random-feature based DrDAM takes $O(D(YK+L(Y+D)))$ time and $O(Y+D)$ peak memory for $L$ energy gradient descent steps (or layers) as defined in eq. (5) with the random feature based approximation gradient $\nabla_{\mathbf{x}}\hat{E}(\mathbf{x})$ defined in eq. (10).*

---

**Algorithm 1:** Procedures for DrDAM with random features.

**RF(Seed $\tau$, Memory $\boldsymbol{\xi}$)**
  Initialize $\mathbf{p} \leftarrow 0_Y$
  Set RNG $R$ seed to $\tau$
  **for** $\alpha = 1,\ldots,Y$ **do**
    Sample random feature $\varphi_\alpha$ from $R$
    $p_\alpha \leftarrow \varphi_\alpha(\boldsymbol{\xi})$
  **return p**

**ProcMems(Seed $\tau$, Mems $\{\boldsymbol{\xi}^\mu, \mu \in \{1,\ldots,K\}\}$)**
  Initialize $\mathbf{T} \leftarrow 0_Y$
  **for** $\mu = 1,\ldots,K$ **do**
    $\mathbf{T} \leftarrow \mathbf{T}+\text{RF}(\tau, \boldsymbol{\xi}^\mu)$
  **return T**

**GradComp(Seed $\tau$, Distributed mems T, Input x)**
  $\mathbf{p} \leftarrow \text{RF}(\tau, \mathbf{g}(\mathbf{x}))$
  Initialize $\mathbf{z} \leftarrow 0_D$
  // Compute $\nabla_{\mathbf{y}}\boldsymbol{\varphi}(\mathbf{y})^\top\mathbf{T}$ in $O(Y)$ mem
  **for** $i = 1,\ldots,D$ **do**
    Compute $\mathbf{u} \leftarrow d\boldsymbol{\varphi}(\mathbf{y})/dy_i|_{\mathbf{y}=\mathbf{g}(\mathbf{x})}$
    $z_i \leftarrow \langle\mathbf{u}, \mathbf{T}\rangle$
  Initialize $\mathbf{z}' \leftarrow 0_D$
  // Compute $\mathbf{z}\nabla_{\mathbf{x}}\mathbf{g}(\mathbf{x})$ in $O(D)$ mem
  **for** $i = 1,\ldots,D$ **do**
    Compute $\mathbf{y} \leftarrow d\mathbf{g}(\mathbf{x})/dx_i$
    $z_i' \leftarrow \langle\mathbf{y}, \mathbf{z}\rangle$
  Compute $q \leftarrow -dQ(s)/ds|_{s=\langle\mathbf{T},\mathbf{p}\rangle}$
  **return** $q\mathbf{z}'$

---

However, note that the memory encoding only needs to be done once, while the same $\mathbf{T}$ can be utilized for $L$ steps of energy gradient steps for multiple input, and the cost of ProcMems is amortized over these multiple inputs. We also show that the computational costs of the inclusion of a new memories $\boldsymbol{\xi}$:

**Proposition 5.** *The inclusion of a new memory $\boldsymbol{\xi} \in \mathbb{R}^D$ to a DrDAM with $K$ memories distributed in $\mathbf{T} \in \mathbb{R}^Y$ takes $O(DY)$ time and $O(D+Y)$ peak memory.*

The above result shows that inclusion of new memories correspond to constant time and memory irrespective of the number of memories in the current DenseAM. Next, we study the divergence between the output of a $L$-layered MrDAM using the energy descent in eq. (5) with the true gradient in eq. (6) and that of DrDAM using the random feature based gradient in eq. (10).

---

[6]If $Q(\cdot) = \log(\cdot)$ in eq. (9), note that the inner product between unconstrained choices of $\boldsymbol{\varphi}$ can be negative but the argument to log must not be; thus, we clip the value to the log to some small $\varepsilon > 0$.

**Theorem 2.** *Consider the following energy function with $K$ memories $\{\boldsymbol{\xi}^\mu \in \mathbb{R}^D, \mu \in \{1, \ldots, K\}\}$ and inverse temperature $\beta > 0$:*

$$E(\mathbf{x}) = -\frac{1}{\beta} \log \left( \sum_{\mu=1}^{K} \exp(-\beta/2 \|\boldsymbol{\xi}^\mu - \mathbf{x}\|_2^2) \right). \tag{11}$$

*We further make the following assumptions: (A1) All memories $\boldsymbol{\xi}^\mu$ and inputs $\mathbf{x}$ lie in $\mathcal{X} = [0, 1/\sqrt{D}]^D$. (A2) Using a random feature map $\boldsymbol{\varphi} : \mathbb{R}^D \to \mathbb{R}^Y$ using $Y$ random feature maps, for any $\mathbf{x}, \mathbf{x}' \in \mathbb{R}^d$ there is a constant $C_1 > 0$ such that $\left| \exp(\|\mathbf{x} - \mathbf{x}'\|_2^2 / 2) - \langle \boldsymbol{\varphi}(\mathbf{x}), \boldsymbol{\varphi}(\mathbf{x}') \rangle \right| \leq C_1 \sqrt{D/Y}$. Given an input $\mathbf{x} \in \mathcal{X}$, let $\mathbf{x}^{(L)}$ be the output of the MrDAM defined by the energy function in eq. (11) using the true energy gradient in eq. (6) and $\hat{\mathbf{x}}^{(L)}$ be the output of DrDAM with approximate gradient in eq. (10) using the random feature map $\boldsymbol{\varphi}$ using a constant step-size of $\eta > 0$ in (5). Then*

$$\left\| \mathbf{x}^{(L)} - \hat{\mathbf{x}}^{(L)} \right\|_2 \leq 2\eta L C_1 K e^{\beta E(\mathbf{x})} \sqrt{D/Y} \left( \frac{1 - \left(\eta L (1 + 2K\beta e^{\beta/2})\right)^L}{1 - \eta L (1 + 2K\beta e^{\beta/2})} \right) \tag{12}$$

Assumption (A1) just ensures that all the memories and inputs have bounded norm, and can be achieved via translating and scaling the memories and inputs. Assumption (A2) pertains to the approximation introduced in the kernel function evaluation with the random feature map, and is satisfied (with high probability) based on results such as Rahimi and Recht [37, Claim 1] and Choromanski et al. [18, Theorem 4]. The above result precisely characterizes the effect on the divergence $\|\mathbf{x}^{(L)} - \hat{\mathbf{x}}^{(L)}\|$ of the (i) initial energy of the input $E(\mathbf{x})$ – lower is better, (ii) the inverse temperature $\beta$ – lower is better, (iii) the number of memories $K$ – lower is better, (iv) the ambient data dimensionality $D$ – lower is better, (v) the number of random features $Y$ – higher is better, and (vi) the number of layers $L$ – lower is better. The proof and further discussion are provided in appendix F.2. Note that theorem 2 analyzes the discretized system, but as the step-size $\eta \to 0$, we approach the fully contracting continuous model. An appropriate choice for the energy descent step-size $\eta$ simplifies the above result, bounding the divergence to $O(\sqrt{D/Y})$:

**Corollary 1.** *Under the conditions and definitions of theorem 2, if we set the step size $\eta = \frac{C_2}{L(1+2K\beta e^{\beta/2})}$ with $C_2 < 1$, the divergence is bounded as:*

$$\left\| \mathbf{x}^{(L)} - \hat{\mathbf{x}}^{(L)} \right\|_2 \leq \frac{C_1 C_2 e^{\beta(E(\mathbf{x}) - 1/2)}}{\beta(1 - C_2)} \sqrt{D/Y}. \tag{13}$$

These above results can be extended to the EDP based energy function $E(\mathbf{x}) = -1/\beta \log \sum_\mu \exp(\beta \langle \boldsymbol{\xi}^\mu, \mathbf{x} \rangle) + 1/2 \|\mathbf{x}\|_2^2$ using the same proof technique.

## 4 Empirical evaluation

To be an accurate approximation of the traditional MrDAM, DrDAM must empirically satisfy the following desiderata for all possible queries and at all configurations for inverse temperature $\beta$ and pattern dimension $D$:

    **(D1)** for the same query, DrDAM must predict similar energies and energy gradients as MrDAM; and

    **(D2)** for the same initial query, DrDAM must retrieve similar fixed points as MrDAM.

However, in our experiments we observed that the approximation quality of DrDAM is strongly affected by the choice of $\beta$ and that the approximation quality decreases the further the query patterns are from the stored memory patterns, as predicted by theorem 2. We characterize this behavior in the following experiments using the trigonometric "SinCos" basis function, which performed best in our ablation experiments (see appendix C), but note that the choice of the random features do play a significant role in the interpretations of these results.

### 4.1 (D1) How accurate are the energies and gradients of DrDAM?

Figure 3 evaluates how well DrDAM, configured at different feature sizes $Y$, approximates the energy and energy gradients of MrDAM configured with different inverse temperatures $\beta$ and storing random binary patterns of dimension $D$. The experimental setup is described below.

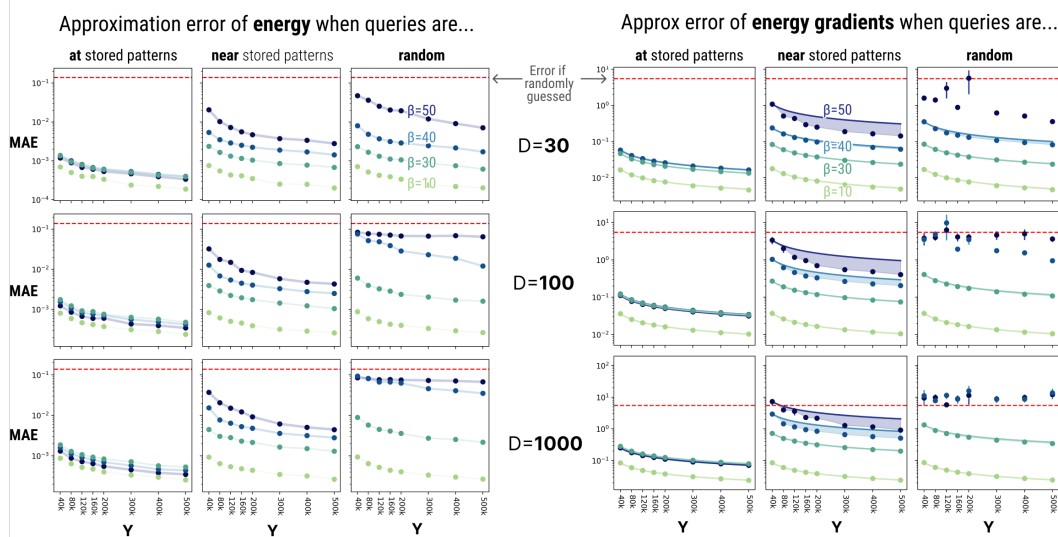

Figure 3: DrDAM produces better approximations to the energies and gradients of MrDAM when the queries are closer to the stored patterns. Approximation quality improves with larger feature dimension $Y$, but decreases with higher $\beta$ and higher pattern dimension $D$. Approximation error is computed on 500 stored binary patterns normalized between $\{0, \frac{1}{\sqrt{D}}\}$. The **M**ean **A**pproximation **E**rrors (**MAE**, eq. (14)) is taken over 500 queries initialized: **at** stored patterns (i.e., queries equal the stored patterns), **near** stored patterns (i.e., queries equal the stored patterns where 10% of the bits have been flipped), and **randomly** (i.e., queries are random and **far** from stored patterns). Error bars represent the standard error of the mean but are visible only at poor approximations. Red horizontal lines represent the expected error of random energies and gradients. The theoretical error upper bounds of eq. (13) (dark curves on the gradient errors in the *right plot only*) show a tight fit to empirical results at low $\beta$ and $D$ and are only shown if predictions are "better than random". The shaded area shows the difference between the theoretical bound and the empirical results.

We generated $2K = 1000$ unique, binary patterns (where each value is normalized to be $\{0, \frac{1}{\sqrt{D}}\}$) and stored $K = 500$ of them into the memory matrix $\Xi$ of MrDAM. We denote these stored patterns as $\boldsymbol{\xi}^{\mu} \in \{0, \frac{1}{\sqrt{D}}\}^{D}$, $\mu \in [\![K]\!]$, where $D$ is a hyperparameter controlled by the experiment. For a given $\beta$, the memory matrix is converted into the featurized memory vector $T_{\alpha} := \sum_{\mu} \varphi_{\alpha}(\boldsymbol{\xi}^{\mu})$ from eq. (9), where $\alpha \in [\![2Y]\!]$. The remaining patterns are treated as the "random queries" $\mathbf{x}_{\text{far}}^{b}$, $b \in [\![K]\!]$ (i.e., queries that are far from the stored patterns). Finally, in addition to evaluating the energy at these random queries and at the stored patterns, we also want to evaluate the energy at queries $\mathbf{x}_{\text{near}}^{b}$ that are "near" the stored patterns; thus, we take each stored pattern $\boldsymbol{\xi}^{\mu}$ and perform bit-flips on $0.1D$ of its entries.

For each set of queries $\mathbf{x}^{b} \in \{\boldsymbol{\xi}^{b}, \mathbf{x}_{\text{near}}^{b}, \mathbf{x}_{\text{far}}^{b}\}$, $b \in [\![K]\!]$, and choice of $\beta$, $Y$, and $D$, we compute the **M**ean **A**pproximation **E**rror (MAE) between MrDAM's energy $E_{b} := E(\mathbf{x}^{b}; \beta, \Xi)$ (whose gradient matrix is denoted $\nabla_{\mathbf{x}} E_{b}$) and DrDAM's energy $\hat{E}_{b} := \hat{E}(\mathbf{x}^{b}; \beta, \mathbf{T})$ (whose gradient is denoted $\nabla_{\mathbf{x}} \hat{E}_{b}$).

$$\text{MAE}_{\text{Energy}} = \frac{1}{K} \sum_{b \in [\![K]\!]} \left| E_{b} - \hat{E}_{b} \right|, \text{ and } \text{MAE}_{\text{Gradient}} = \frac{1}{K} \sum_{b \in [\![K]\!]} \left\| \nabla_{\mathbf{x}} E_{b} - \nabla_{\mathbf{x}} \hat{E}_{b} \right\|_{2} \quad (14)$$

We found it useful to visualize the results using log-scale and to compare the errors against the expected error of a "random guess" of the energy/gradients (horizontal red dashed line in each plot of fig. 3). The "random guess error" was empirically computed by sampling a new set of random queries $\mathbf{x}_{\text{guess}}^{b}$, $b \in [\![K]\!]$ (independent of the reference queries) and computing the MAE between the standard energy on the reference queries vs. the approximate energies on the random queries. This error was averaged across $Y$ for each $\beta$; the highest average error across all $\beta$s is plotted.

**Observation 1: DrDAM approximations are best for queries near stored patterns** DrDAM approximations for both the energy and energy gradients are better the closer the query patterns are to the stored patterns. In this regime, approximation accuracy predictably improves when increasing

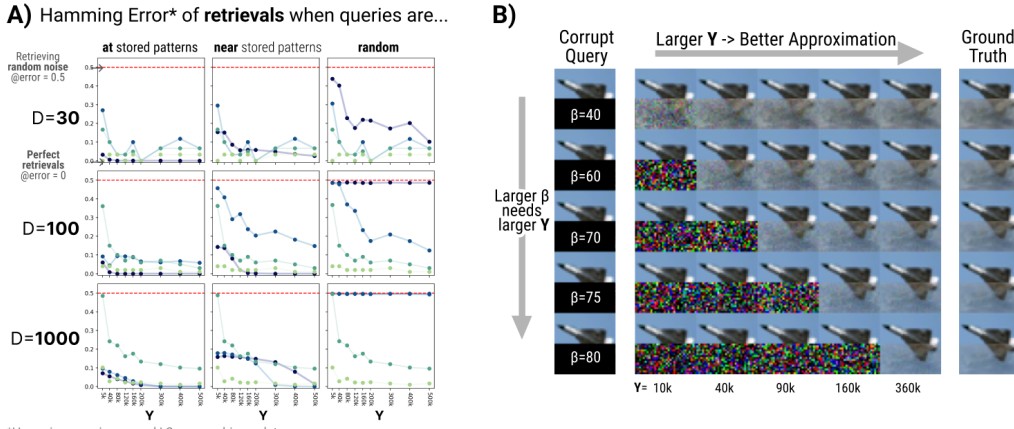

Figure 4: **A)** Retrieval errors predictably follow the approximation quality of fig. 3. Error is lowest at/near stored patterns but is completely random when energy and gradient approximations are poor, i.e., at high values of $\beta$ and $D$. Note that error improves across $Y$ but follows a different (and noisier) trace than the corresponding approximations for energy and gradient in fig. 3 due to error accumulating over multiple update steps. **B)** DrDAM's approximation quality improves as $Y$ increases (visible at low $\beta$), but larger $Y$'s are needed for good approximations to the DAM's fixed points at higher $\beta$'s. (Left) The same corrupted query from CIFAR-10 where bottom 50% is masked is presented to DAM's with different $\beta$'s. (Middle) The fixed points of DrDAM for each $\beta$ at different sizes $Y$ of the feature space. (Right) The "ground truth" fixed point of MrDAM. The top 50% of pixels are clamped throughout the dynamics.

the value for $Y$ within "reasonable" values (i.e., values corresponding into sizes of featurized queries and memories that can operate within 46GB of GPU memory).

**Observation 2: DrDAM approximations worsen as inverse temperature $\beta$ increases**    Across nearly all experiments, DrDAM approximations worsen as $\beta$ increases. At queries near the stored patterns, $\beta = 50$ has an energy error approximately $10\times$ that of $\beta = 30$ and $100\times$ that of $\beta = 10$ across all $Y$. At high $D$ and when queries are far from the patterns, the error of $\beta = 50$ approaches $1000\times$ the error of $\beta = 10$. This observation similarly holds for the errors of corresponding gradients, corroborating the statement of theorem 2.

**Observation 3: DrDAM approximations break at sufficiently high values of $D$ and $\beta$**    In general, DrDAM's approximation errors remain the same across choices for $D$, especially when the queries are near the stored patterns. However, when both $\beta$ and $D$ are sufficiently large (e.g., $\beta \geq 40$ and $D \geq 100$ in fig. 3), increasing the value of $Y$ does not improve the approximation quality: DrDAM continues to return almost random gradients and energies. We explore this phenomenon more in § 4.2 in the context of the retrievability of stored patterns.

## 4.2  (D2) How accurate are the memory retrievals using DrDAM?

*Memory retrieval* is the process by which an initial query $\mathbf{x}^{(0)}$ descends the energy function and is transformed into a fixed point of the energy dynamics. This process can be described by the discrete update rule in eq. (5), where $E$ can represent either MrDAM's energy or the approximate energy of DrDAM. A memory is said to be "retrieved" when $|E(\mathbf{x}^{(L)}) - E(\mathbf{x}^{(L-1)})| < \varepsilon$ for some small $\varepsilon > 0$, at which point $\mathbf{x}^{(L-1)} \approx \mathbf{x}^{(L)} =: \mathbf{x}^\star$ is declared to be the retrieved *memory* after $L$ iterations because $\mathbf{x}^\star$ lives at a local minimum of the energy function $E$.

**Quantifying retrieval error**    Given the same initial queries $\mathbf{x}^{(0)} \in \{0, \frac{1}{\sqrt{D}}\}^D$, we want to quantify the difference between the fixed points $\hat{\mathbf{x}}^\star$ retrieved by descending DrDAM's approximate energy and the fixed points $\mathbf{x}^\star$ retrieved by descending the energy of MrDAM. We follow the experimental setup of § 4.1, only this time we run full memory retrieval dynamics until convergence.

Note that since energy uses an L2-similarity kernel, memory retrieval is not guaranteed to return binary values. Thus, we binarize $\mathbf{x}^\star$ by assigning each entry to its nearest binary value before computing the normalized Hamming approximation error $\Delta_H$, i.e.,

$$\lceil x \rceil := \begin{cases} \dfrac{1}{\sqrt{D}}, & x \geq \dfrac{1}{2\sqrt{D}} \\ 0, & \text{otherwise} \end{cases} \text{, and} \qquad \Delta_H := \frac{1}{\sqrt{D}} \sum_{i \in [\![D]\!]} \left| \lceil \mathbf{x}_i^\star \rceil - \lceil \hat{\mathbf{x}}_i^\star \rceil \right|. \tag{15}$$

The choice of normalized Hamming approximation error $\Delta_H$ on our binary data is equivalent to the squared L2 error on the left side of our bound in eq. (13) (up to a linear scaling of $\frac{1}{\sqrt{D}}$).

Figure 4A shows the results of this experiment. Many observations from § 4.1 translate to these experiments: we notice that retrieval is random at high $\beta$ and $D$, and that retrievals are of generally higher accuracy nearer the stored patterns. However, we notice that high $\beta$ values can retrieve better approximations than lower values of $\beta$ when the queries are at or near stored patterns. Additionally, for sufficiently high $\beta$ (e.g., see $D = 1000$, $\beta = 50$ near stored patterns), this accompanies an interesting "thresholding" behavior for $Y$ where retrieval error starts to improve rapidly once $Y$ reaches a minimal threshold. This behavior is corroborated in the high $D$ regime in fig. 4B.

**Visualizing retrieval error**    Figure 4B shows what retrieval errors look like qualitatively. We stored $K = 10$ random images from CIFAR10 [43] into the memory matrix of MrDAM, resulting in patterns of size $D = 3 \times 32 \times 32 = 3072$, and compared retrievals using $\beta$s that produced meaningful image results with MrDAM. To keep $\beta$ values consistent with our previous experiments, each pixel was normalized to the continuous range between 0 and $\frac{1}{\sqrt{D}}$ s.t. $\xi_i^\mu \in [0, \frac{1}{\sqrt{D}}]$, with $\mu \in [\![K]\!]$ and $i \in [\![D]\!]$.

From § 4.1 and fig. 4A, we know that approximate retrievals are inaccurate at high $\beta$ and high $D$ if the query is far from the stored patterns. However, this is exactly the regime we test when retrieving images in fig. 4B. The visible pixels (top half of the image) are clamped while running the dynamics until convergence. Retrieved memories at different configurations for DrDAM are plotted against their corresponding MrDAM retrievals in fig. 4B.

As $\beta$ increases, insufficiently large values of $Y$ fail to retrieve meaningful approximations to the dynamics of MrDAM. We observe that image completions generally become less noisy as $Y$ increases, but with diminishing improvement in perceptible quality after some threshold where DrDAM goes from predicting noise to predicting meaningful image completions.

## 5    Conclusion

Our study is explicitly designed to characterize where DrDAM is a good approximation to the energies and dynamics of MrDAM. In pushing the limits of the distributed representation, we discovered that DrDAM is most accurate when: (1) query patterns are nearer to the stored patterns; (2) $\beta$ is lower; and (3) $Y$ is large. Error bounds for these situations are explicitly derived in theorem 2 and empirically tested in § 4.

We have explored the use of distributed representations via random feature maps in DenseAMs. We have demonstrated how this can be done efficiently, and we precisely characterized how it performs the neural dynamics relative to the memory representation DenseAMs. Our theoretical results highlight the factors playing a role in the approximation introduced by the distributed representations, and our experiments validate these theoretical insights. As future work, we intend to explore how such distributed representations can be leveraged in hierarchical associative memory networks [44, 45], which can have useful inductive biases (e.g., convolutions, attention), and allow extensions with multiple hidden layers.

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

# A Limitations

In this paper, we have explored the use of distributed representations via random feature maps in DenseAMs. However, we are only scratching the surface of opportunities that such distributed representations bring to DenseAMs. There are various aspects we do not cover: (i) We do not cover the ability of these distributed representations to provide (probably lossy) compression. (ii) We do not study the properties of DrDAM relative to MrDAM when DrDAM is allowed to have different step sizes and number of layers than MrDAM. A further limitation of our work is the limited number of datasets on which we have characterized the performance of DrDAM.

# B Approximation error when increasing the number of stored patterns in DrDAM

§ 4.1 validated eq. (13), confirming that approximation error decreases as the number of random features $Y$ increases under constant number of stored patterns $K$. We can also consider a related but different question: under constant number of random features $Y$, how does approximation error behave when increasing the number of stored patterns $K$? Intuitively, DrDAM's approximation should be good when a small number of patterns are stored in the network, and this approximation should worsen as we increase the number of stored patterns.

Figure 5 validates this intuition empirically, with the caveat that random queries generally improve in accuracy because the probability of being near a stored patterns (a regime that generally leads to higher accuracy of retrievals, see § 4) increases as we store more patterns into the network. For this experiment, $Y = 2e5$ was held constant across all experiments and each plotted approximation error is averaged over a number of queries equal to the number of stored patterns $K$. The experimental design otherwise exactly replicates that of fig. 3.

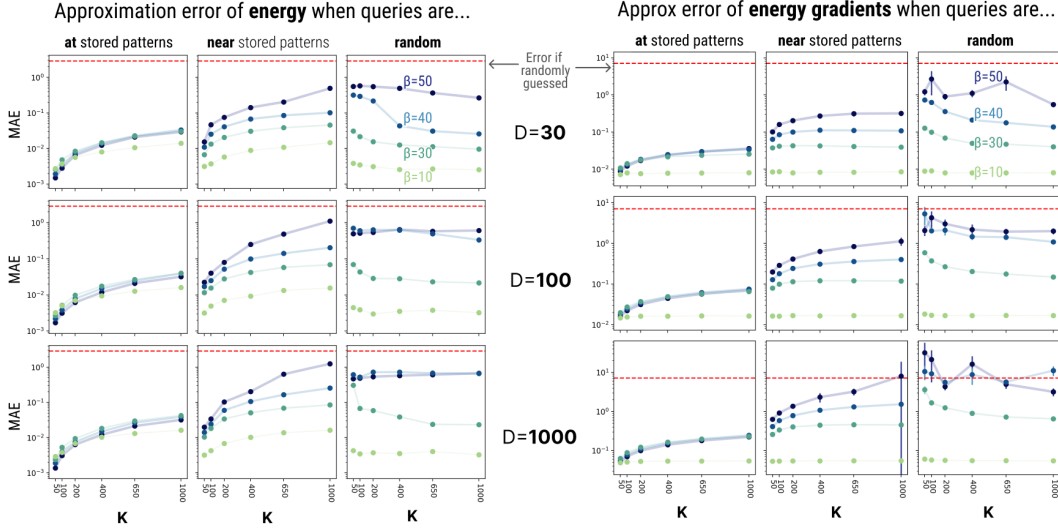

Figure 5: Mean Approximation Error (MAE, eq. (14)) increases as the number of stored patterns $K$ increases (except at random starting positions, where more stored patterns increases the probability that a random query is closer to a memory, a regime that leads to higher accuracy of the retrievals, see fig. 3), keeping $Y = 2e5$ constant across all experiments.

# C Ablation study: Comparing choices for basis function

Different basis functions can be used to approximate the RBF kernel used in the energy of the memory representation of the DAM in eq. (7). We considered the following kernels ("Cos", "SinCos", "Exp", "ExpExp"), rewritten here as

$$\varphi_{\text{Cos}}(\mathbf{x}) \quad = \sqrt{\frac{2}{Y}} \begin{bmatrix} \cos(\langle \boldsymbol{\omega}^1, \mathbf{x} \rangle + b_1) \\ \cos(\langle \boldsymbol{\omega}^2, \mathbf{x} \rangle + b_2) \\ \ldots, \\ \cos(\langle \boldsymbol{\omega}^Y, \mathbf{x} \rangle + b_Y) \end{bmatrix},$$

$$\varphi_{\text{SinCos}}(\mathbf{x}) \quad = \frac{1}{\sqrt{Y}} \begin{bmatrix} \cos(\langle \boldsymbol{\omega}^1, \mathbf{x} \rangle) \\ \sin(\langle \boldsymbol{\omega}^1, \mathbf{x} \rangle) \\ \cos(\langle \boldsymbol{\omega}^2, \mathbf{x} \rangle) \\ \sin(\langle \boldsymbol{\omega}^2, \mathbf{x} \rangle) \\ \ldots, \\ \cos(\langle \boldsymbol{\omega}^Y, \mathbf{x} \rangle) \\ \sin(\langle \boldsymbol{\omega}^Y, \mathbf{x} \rangle) \end{bmatrix},$$

$$\varphi_{\text{Exp}}(\mathbf{x}) \quad = \frac{\exp(-\|\mathbf{x}\|_2^2)}{\sqrt{Y}} \begin{bmatrix} \exp(\langle \boldsymbol{\omega}^1, \mathbf{x} \rangle + b_1) \\ \exp(\langle \boldsymbol{\omega}^2, \mathbf{x} \rangle + b_2) \\ \ldots, \\ \exp(\langle \boldsymbol{\omega}^Y, \mathbf{x} \rangle + b_Y) \end{bmatrix},$$

$$\varphi_{\text{ExpExp}}(\mathbf{x}) = \frac{\exp(-\|\mathbf{x}\|_2^2)}{\sqrt{2Y}} \begin{bmatrix} \exp(+\langle \boldsymbol{\omega}^1, \mathbf{x} \rangle) \\ \exp(-\langle \boldsymbol{\omega}^1, \mathbf{x} \rangle) \\ \exp(+\langle \boldsymbol{\omega}^2, \mathbf{x} \rangle) \\ \exp(-\langle \boldsymbol{\omega}^2, \mathbf{x} \rangle) \\ \ldots, \\ \exp(+\langle \boldsymbol{\omega}^Y, \mathbf{x} \rangle) \\ \exp(-\langle \boldsymbol{\omega}^Y, \mathbf{x} \rangle) \end{bmatrix},$$

where $\boldsymbol{\omega}^\alpha \sim \mathcal{N}(0, I_D)$, $\alpha \in [\![Y]\!]$ are the random projection vectors and $b^\alpha \sim \mathcal{U}(0, 2\pi)$ are random "biases" or shifts in the basis function.

Figure 6 shows how well the above basis functions approximated the true energy and energy gradient at different values for $\beta$ and size of feature dimension $Y$. Specifically, given the Letter dataset [46] which consists of 16-dimensional continuous vectors whose values were normalized to be between $[0, \frac{1}{\sqrt{D}}]$, we randomly selected 900 unique data points, storing 500 patterns into the memory and choosing the remaining 400 to serve as new patterns. We then compared how well the energy and energy gradients of the chosen basis function approximates the predictions of the original DAM.

We observe that the trigonometric basis functions (i.e., either Cos or SinCos) provide the most accurate approximations for the energy and gradients of the standard MrDAM, especially in the regime of high $\beta$ which is required for the high memory storage capacity of DenseAMs. Surprisingly, the Positive Random Features (PRFs) of [18] do not perform well in the dense (high $\beta$) regime; in general, trigonometric features always provide better approximations than the PRFs.

We conclude that the SinCos basis function is the best approximation for use in the experiments reported in the main paper, as this choice consistently produces the best approximations for the energy gradients across all values of $\beta$.

## D  TinyImagenet Experimental Details

In performing the qualitative reconstructions shown in fig. 1, we used a standard MrDAM energy (eq. (7)) configured with inverse temperature $\beta = 60$. We approximated this energy in a DrDAM using the trigonometric "SinCos" basis function shown in eq. (8) configured with feature dimension $Y = 1.8e5$. The four images shown were selected from the Tiny Imagenet [11] dataset, rasterized into a vector, and stored in the memory matrix a MrDAM, resulting in a memory of shape $(4, 12288)$. Energy descent for both MrDAM and DrDAM used standard gradient descent at a step size of 0.1 until the dynamics of all images converged (for fig. 1 after 300 steps, see energy traces). Visible pixels are "clamped" throughout the duration of the dynamics by zeroing out the energy gradients on the visible top one-third of the image.

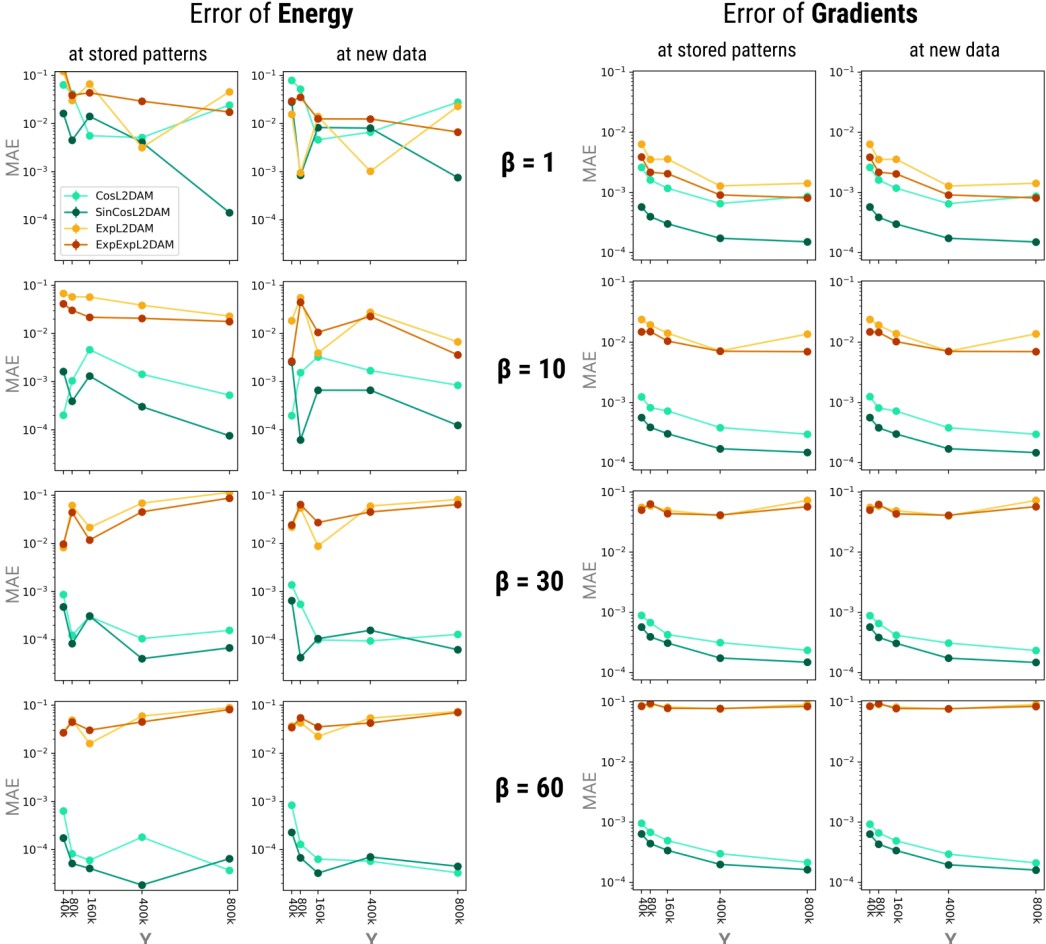

Figure 6: Trigonometric basis functions significantly outperform Positive Random Features, especially in the regime of large $\beta$. We end up choosing the SinCos function to analyze in the main paper, as this choice of basis function always produced the best approximations to the energy gradient. Experiments performed on the 16-dimensional Letters dataset [46].

In MrDAM, the memory matrix necessarily grows linearly when storing new patterns $\boldsymbol{\xi}^\mu$. However, the distributed memory tensor $\mathbf{T}$ of DrDAM does not grow when new patterns are stored. This means it is possible to compress the memories into a smaller tensor $\mathbf{T}$ where $Y <$ NumPixels, provided that we operate in a regime that allows larger approximation errors in the retrieval and smaller initial occlusions. Figure 2 shows a variation of the setting of fig. 1 where stored patterns are actually *compressed* into DrDAM's memory tensor, successfully storing $20 * 12288$ pixels from a distributed tensor of size $Y = 2e5$ and retrieving the memories with $40\%$ initial occlusion of the queries, a $\sim 20\%$ reduction in the number of parameters compared to MrDAM. All other hyperparameters are the same as was used to generate fig. 1, and convergence on all images occurs after $1000$ steps.

## E   Details on Computational Environment for the Experiments

All experiments are performed on a single L40s GPU equiped with 46GB VRAM. Experiments were written and performed using the JAX [47] library for tensor manipulations. Unless otherwise noted, gradient computation was performed using JAX's powerful autograd mechanism. Experimental code with instructions to replicate the results in this paper are made available at this GitHub repository (`https://github.com/bhoov/distributed_DAM`), complete with instructions to setup the coding environment and run all experiments.

## F Detailed Proofs and Discussions

### F.1 Details for theorem 1

#### F.1.1 Proof of theorem 1

*Proof of theorem 1.* The proof above involves noting that, first we need to encode all the memories with ProcMems, which takes $O(DYK)$ time and $O(Y + D)$ peak memory using proposition 3.

Then we compute $L$ gradients with GradComp for $L$ iterations of energy gradient descent, taking $O(LD(Y + D))$ time and $O(D + Y)$ peak memory using proposition 4.

Putting the runtimes together, and using the maximum of the peak memories gives us the statement of the theorem. $\qquad\square$

#### F.1.2 Comparing computational complexities of MrDAM and DrDAM

Note that, comparing the computational complexities of MrDAM in proposition 1 to that of DrDAM in theorem 1 does not directly provide any computational improvements as it would depend on the choices of $D, K, L, Y$. The main point of these results is to highlight, that once the memories are processed via ProcMems, the energy descent with DrDAM requires computation and memory that only depends on $D$ and $Y$. And together with theorem 2 and corollary 1, we characterize situations where the energy descent divergence between MrDAM and DrDAM can be bounded with a choice of $Y$ that only depends on $D$ (and other parameters in the energy function) but not $K$.

While we do not claim or highlight computational gains over MrDAM, note that the peak memory complexity of MrDAM is $O(KD)$ compared to $O(Y + D)$ for DrDAM. Given that in the interesting regime of $Y \sim O(D/\epsilon^2)$ which upperbounds the energy descent divergence between DrDAM and MrDAM in corollary 1 to at most some $\epsilon > 0$, DrDAM is more memory efficient than MrDAM if the number of memories $K > C/\epsilon^2$ for some sufficiently large positive constant $C$. Ignoring the time required to encode the memories into the distributed representation in DrDAM using ProcMems, the runtime complexities are $O(LKD)$ for MrDAM compared to $O(LD(Y + D))$ for DrDAM. Again, considering the interesting regime of $Y \sim O(D/\epsilon^2)$, DrDAM will be computationally more efficient than MrDAM if the number of memories $K > \widetilde{C}D/\epsilon^2$ for some sufficiently large positive constant $\widetilde{C}$.

### F.2 Details for theorem 2

#### F.2.1 Proof of theorem 2

Here we will make use of the following result from Li et al. [48]:

**Lemma 1** (adapted from Li et al. [48] Lemma B.1)**.** *For* $\mathbf{x}, \mathbf{z} \in \mathbb{R}^K$ *with* $\max_{i,j \in [\![K]\!]}(\mathbf{x}_i - \mathbf{x}_j) \leq \delta$ *and* $\max_{i,j \in [\![K]\!]}(\mathbf{z}_i - \mathbf{z}_j) \leq \delta$, *we have the following:*

$$\|\texttt{softmax}(\mathbf{x})\|_\infty \leq \frac{e^\delta}{K}, \quad \|\texttt{softmax}(\mathbf{x}) - \texttt{softmax}(\mathbf{z})\|_1 \leq \frac{e^\delta}{K}\|\mathbf{x} - \mathbf{z}\|_1. \tag{16}$$

We now develop the following results:

**Lemma 2.** *Under the conditions and notation of theorem 2, for* $\mathbf{x}, \mathbf{z} \in \mathcal{X}$, *we have*

$$\|\nabla_\mathbf{x} E(\mathbf{x}) - \nabla_\mathbf{x} E(\mathbf{z})\| \leq (1 + 2K\beta e^{\beta/2})\|\mathbf{x} - \mathbf{z}\|. \tag{17}$$

*Proof.* Given the energy function in eq. (11), we can write the energy gradient $\nabla_\mathbf{x} E(\mathbf{x})$ as:

$$\nabla_\mathbf{x} E(\mathbf{x}) = \texttt{softmax}(-\beta/2\|\mathbf{x} - \Xi\|_2^2)(\mathbf{x} - \Xi) = \mathbf{x} - \texttt{softmax}(-\beta/2\|\mathbf{x} - \Xi\|_2^2)\Xi, \tag{18}$$

where $\Xi = [\xi^1, \ldots, \xi^K]$, $\|\mathbf{x} - \Xi\|_2^2$ denotes $[\|\mathbf{x} - \xi^1\|_2^2, \ldots \|\mathbf{x} - \xi^K\|_2^2]$ and $(\mathbf{x} - \Xi)$ denotes $[(\mathbf{x} - \xi^1), \ldots, (\mathbf{x} - \xi^K)]$. Then we have

$$\|\nabla_{\mathbf{x}} E(\mathbf{x}) - \nabla_{\mathbf{x}} E(\mathbf{z})\|_2 \tag{19}$$

$$= \|\mathbf{x} - \mathtt{softmax}(-\beta/2\|\mathbf{x} - \Xi\|_2^2)\Xi - \mathbf{z} + \mathtt{softmax}(-\beta/2\|\mathbf{z} - \Xi\|_2^2)\Xi\|_2 \tag{20}$$

$$\leq \|\mathbf{x} - \mathbf{z}\| + \|(\mathtt{softmax}(-\beta/2\|\mathbf{x} - \Xi\|_2^2) - \mathtt{softmax}(-\beta/2\|\mathbf{z} - \Xi\|_2^2))\Xi\|_2 \tag{21}$$

$$\leq \|\mathbf{x} - \mathbf{z}\| + \|(\mathtt{softmax}(-\beta/2\|\mathbf{x} - \Xi\|_2^2) - \mathtt{softmax}(-\beta/2\|\mathbf{z} - \Xi\|_2^2))\|_1 \|\Xi\|_2 \tag{22}$$

$$\leq \|\mathbf{x} - \mathbf{z}\| + \frac{e^{\beta/2}}{K}\|\Xi\|_2 \left\|\beta/2(\|\mathbf{z} - \Xi\|_2^2 - \|\mathbf{x} - \Xi\|_2^2)\right\|_1, \tag{23}$$

where we applied lemma 1 to the softmax term in the right hand side of eq. (22) with $\delta = \beta/2$ since all pairwise distances in $\mathcal{X}$ are in $[0, 1]$.

Now we have

$$\left\|\beta/2(\|\mathbf{z} - \Xi\|_2^2 - \|\mathbf{x} - \Xi\|_2^2)\right\|_1 = \frac{\beta}{2}\sum_{\mu=1}^{K}\left|\|\mathbf{z} - \boldsymbol{\xi}^\mu\|_2^2 - \|\mathbf{x} - \boldsymbol{\xi}^\mu\|_2^2\right| \tag{24}$$

$$= \frac{\beta}{2}\sum_{\mu=1}^{K}|\langle\mathbf{z} + \mathbf{x}, \mathbf{z} - \mathbf{x}\rangle + 2\langle\boldsymbol{\xi}^\mu, \mathbf{x} - \mathbf{z}\rangle| \tag{25}$$

$$\leq \frac{\beta}{2}\sum_{\mu=1}^{K}\|\mathbf{z} - \mathbf{x}\|(\|\mathbf{z} + \mathbf{x}\| + 2\|\boldsymbol{\xi}^\mu\|) \leq \frac{\beta}{2}\sum_{\mu=1}^{K}4\|\mathbf{z} - \mathbf{x}\|, \tag{26}$$

since $\|\boldsymbol{\xi}^\mu\| \leq 1$ and $\|\mathbf{x} + \mathbf{z}\| \leq \|\mathbf{x}\| + \|\mathbf{z}\| \leq 2$. Putting eq. (26) in eq. (23), and using the fact that $\|\Xi\|_2 \leq K$, we have

$$\|\nabla_{\mathbf{x}} E(\mathbf{x}) - \nabla_{\mathbf{x}} E(\mathbf{z})\|_2 \leq \|\mathbf{x} - \mathbf{z}\| + \frac{e^{\beta/2}}{K}K2\beta\sum_{\mu=1}^{K}\|\mathbf{z} - \mathbf{x}\| = (1 + 2K\beta e^{\beta/2})\|\mathbf{x} - \mathbf{z}\|, \tag{27}$$

giving is eq. (16) in the statement of the lemma. $\qquad\square$

Given the structure of the energy gradient in eq. (18) of the energy function in eq. (11), we consider a specific energy gradient for this specific energy function instead of the generic energy gradient in eq. (10). We can rewrite the exact energy gradient as

$$\nabla_{\mathbf{x}} E(\mathbf{x}) = \mathbf{x} - \sum_{\mu=1}^{K}\frac{\exp(-\beta/2\|\mathbf{x} - \boldsymbol{\xi}^\mu\|_2^2)\boldsymbol{\xi}^\mu}{\sum_{\mu'=1}^{K}\exp(-\beta/2\|\mathbf{x} - \boldsymbol{\xi}^{\mu'}\|_2^2)}. \tag{28}$$

Using random feature maps, we can write the approximate gradient as

$$\nabla_{\mathbf{x}} \hat{E}(\mathbf{x}) = \mathbf{x} - \sum_{\mu=1}^{K}\frac{\left\langle\boldsymbol{\varphi}(\sqrt{\beta}\mathbf{x}), \boldsymbol{\varphi}(\sqrt{\beta}\boldsymbol{\xi}^\mu)\right\rangle\boldsymbol{\xi}^\mu}{\sum_{\mu'=1}^{K}\left\langle\boldsymbol{\varphi}(\sqrt{\beta}\mathbf{x}), \boldsymbol{\varphi}(\sqrt{\beta}\boldsymbol{\xi}^{\mu'})\right\rangle} \tag{29}$$

$$= \frac{\sum_{\mu=1}^{K}\boldsymbol{\varphi}(\sqrt{\beta}\mathbf{x}) \cdot \boldsymbol{\varphi}(\sqrt{\beta}\boldsymbol{\xi}^\mu) \cdot \boldsymbol{\xi}^{\mu\top}}{\left\langle\boldsymbol{\varphi}(\sqrt{\beta}\mathbf{x}), \sum_{\mu'=1}^{K}\boldsymbol{\varphi}(\sqrt{\beta}\boldsymbol{\xi}^{\mu'})\right\rangle} \tag{30}$$

$$= \frac{\boldsymbol{\varphi}(\sqrt{\beta}\mathbf{x}) \cdot \sum_{\mu=1}^{K}\boldsymbol{\varphi}(\sqrt{\beta}\boldsymbol{\xi}^\mu) \cdot \boldsymbol{\xi}^{\mu\top}}{\left\langle\boldsymbol{\varphi}(\sqrt{\beta}\mathbf{x}), \mathbf{T}\right\rangle}, \quad \text{where } \mathbf{T} = \sum_{\mu'=1}^{K}\boldsymbol{\varphi}(\sqrt{\beta}\boldsymbol{\xi}^{\mu'}) \tag{31}$$

$$= \frac{\boldsymbol{\varphi}(\sqrt{\beta}\mathbf{x}) \cdot \mathbf{R}}{\left\langle\boldsymbol{\varphi}(\sqrt{\beta}\mathbf{x}), \mathbf{T}\right\rangle}, \quad \text{where } \mathbf{R} = \sum_{\mu=1}^{K}\boldsymbol{\varphi}(\sqrt{\beta}\boldsymbol{\xi}^\mu) \cdot \boldsymbol{\xi}^{\mu\top}, \tag{32}$$

where we again just need to store $\mathbf{T}$ and $\mathbf{R}$ as defined above and do not need to maintain the original memory matrix $\Xi$.

**Lemma 3.** *Under the conditions and notation of theorem 2, and assuming that $\langle \varphi(\mathbf{x}), \varphi(\mathbf{x}') \rangle \geq 0 \forall \mathbf{x}, \mathbf{x}' \in \mathcal{X}$, for the approximate gradient $\nabla_{\mathbf{x}} \hat{E}(\mathbf{x})$ in eq. (32), we have*

$$\|\nabla_{\mathbf{x}} E(\mathbf{x}) - \nabla_{\mathbf{x}} \hat{E}(\mathbf{x})\| \leq 2C_1 K e^{\beta E(\mathbf{x})} \sqrt{\frac{D}{Y}}. \tag{33}$$

*Proof.* We can expand out the left-hand side of eq. (33) as follows:

$$\|\nabla_{\mathbf{x}} E(\mathbf{x}) - \nabla_{\mathbf{x}} \hat{E}(\mathbf{x})\| = \left\| \frac{\sum_{\mu=1}^{K} \exp(-\beta/2\|\mathbf{x} - \boldsymbol{\xi}^\mu\|^2) \boldsymbol{\xi}^\mu}{\sum_{\mu=1}^{K} \exp(-\beta/2\|\mathbf{x} - \boldsymbol{\xi}^\mu\|^2)} - \frac{\sum_{\mu=1}^{K} \langle \varphi(\sqrt{\beta}\mathbf{x}), \varphi(\sqrt{\beta}\boldsymbol{\xi}^\mu) \rangle \boldsymbol{\xi}^\mu}{\sum_{\mu=1}^{K} \langle \varphi(\sqrt{\beta}q), \varphi(\sqrt{\beta}\boldsymbol{\xi}^\mu) \rangle} \right\| \tag{34}$$

by reversing the simplifying steps made above to arrive at eq. (32).

Let use denote $\epsilon = C_1 \sqrt{D/Y}$ the approximation in the kernel value induced by the random feature map $\varphi$. Then considering the terms in the denominator above, we have

$$(1/K) \left| \sum_\mu \exp(-\beta/2\|\mathbf{x} - \boldsymbol{\xi}^\mu\|^2) - \left\langle \varphi(\sqrt{\beta}\mathbf{x}), \sum_\mu \varphi(\sqrt{\beta}\boldsymbol{\xi}^\mu) \right\rangle \right| \tag{35}$$

$$= (1/K) \left| \sum_\mu \left( \exp(-\beta/2\|\mathbf{x} - \boldsymbol{\xi}^\mu\|^2) - \left\langle \varphi(\sqrt{\beta}\mathbf{x}), \varphi(\sqrt{\beta}\boldsymbol{\xi}^\mu) \right\rangle \right) \right| \tag{36}$$

$$\leq (1/K) \sum_\mu \left| \left( \exp(-\beta/2\|\mathbf{x} - \boldsymbol{\xi}^\mu\|^2) - \left\langle \varphi(\sqrt{\beta}\mathbf{x}), \varphi(\sqrt{\beta}\boldsymbol{\xi}^\mu) \right\rangle \right) \right| \leq (1/K) \sum_\mu \epsilon = \epsilon. \tag{37}$$

Considering the terms in the numerators, we have

$$(1/K) \left\| \sum_\mu \exp(-\beta/2\|\mathbf{x} - \boldsymbol{\xi}^\mu\|^2) \boldsymbol{\xi}^\mu - \left\langle \varphi(\sqrt{\beta}\mathbf{x}), \sum_\mu \varphi(\sqrt{\beta}\boldsymbol{\xi}^\mu) \right\rangle \boldsymbol{\xi}^\mu \right\| \tag{38}$$

$$= (1/K) \left\| \sum_\mu \left( \exp(-\beta/2\|\mathbf{x} - \boldsymbol{\xi}^\mu\|^2) - \left\langle \varphi(\sqrt{\beta}\mathbf{x}), \varphi(\sqrt{\beta}\boldsymbol{\xi}^\mu) \right\rangle \right) \boldsymbol{\xi}^\mu \right\| \tag{39}$$

$$\leq (1/K) \sum_\mu \left\| \left( \exp(-\beta/2\|\mathbf{x} - \boldsymbol{\xi}^\mu\|^2) - \left\langle \varphi(\sqrt{\beta}\mathbf{x}), \varphi(\sqrt{\beta}\boldsymbol{\xi}^\mu) \right\rangle \right) \boldsymbol{\xi}^\mu \right\| \tag{40}$$

$$\leq (1/K) \sum_\mu \left| \left( \exp(-\beta/2\|\mathbf{x} - \boldsymbol{\xi}^\mu\|^2) - \left\langle \varphi(\sqrt{\beta}\mathbf{x}), \varphi(\sqrt{\beta}\boldsymbol{\xi}^\mu) \right\rangle \right) \right| \|\boldsymbol{\xi}^\mu\| \tag{41}$$

$$\leq (1/K) \sum_\mu \epsilon \|\boldsymbol{\xi}^\mu\| = \epsilon \qquad \because \|\boldsymbol{\xi}^\mu\| \leq 1. \tag{42}$$

Let us define the following terms for convenience:
- $a = 1/K \sum_\mu \exp(-\beta/2\|\mathbf{x} - \boldsymbol{\xi}^\mu\|^2) \boldsymbol{\xi}^\mu$
- $b = 1/K \sum_\mu \exp(-\beta/2\|\mathbf{x} - \boldsymbol{\xi}^\mu\|^2)$
- $\hat{a} = 1/K \sum_\mu \langle \varphi(\sqrt{\beta}\mathbf{x}), \varphi(\sqrt{\beta}\boldsymbol{\xi}^\mu) \rangle \boldsymbol{\xi}^\mu$
- $\hat{b} = 1/K \sum_\mu \langle \varphi(\sqrt{\beta}\mathbf{x}), \varphi(\sqrt{\beta}\boldsymbol{\xi}^\mu) \rangle$

Then, based on our previous bounds, we know that

$$\|a - \hat{a}\| \leq \epsilon, \quad |b - \hat{b}| \leq \epsilon, \quad \|\nabla_{\mathbf{x}} E(\mathbf{x}) - \nabla_{\mathbf{x}} \hat{E}(\mathbf{x})\| = \left\| \frac{a}{b} - \frac{\hat{a}}{\hat{b}} \right\| \tag{43}$$

$$\|\nabla_{\mathbf{x}}E(\mathbf{x}) - \nabla_{\mathbf{x}}\hat{E}(\mathbf{x})\| = \left\|\frac{a}{b} - \frac{\hat{a}}{\hat{b}}\right\| = \left\|\frac{a - \hat{a}}{b} + \frac{\hat{a}}{\hat{b}}\frac{\hat{b}}{b} - \frac{\hat{a}}{\hat{b}}\right\| \leq \left\|\frac{a - \hat{a}}{b}\right\| + \left\|\frac{\hat{a}}{\hat{b}}\right\|\left|\frac{\hat{b}}{b} - 1\right| \quad (44)$$

$$\leq \frac{1}{b}\|a - \hat{a}\| + \left\|\frac{\hat{a}}{\hat{b}}\right\|\frac{1}{b}|\hat{b} - b| \leq \frac{1}{b}\left(\epsilon + \left\|\frac{\hat{a}}{\hat{b}}\right\|\epsilon\right) \quad (45)$$

$$\leq \epsilon\frac{1}{b}\left(1 + \left\|\frac{\hat{a}}{\hat{b}}\right\|\right) \quad (46)$$

.

Note that $(\hat{a}/\hat{b})$ is in the convex hull of the memories since this is a weighted sum of the memories where the weights are positive and add up to 1. Thus within $(\hat{a}/\hat{b}) \in [0, 1/\sqrt{d}]^d$, thus $\|\hat{a}/\hat{b}\| \leq 1$. Now $b = (1/K)\exp(-\beta E(\mathbf{x}))$. Thus

$$\|\nabla_{\mathbf{x}}E(\mathbf{x}) - \nabla_{\mathbf{x}}\hat{E}(\mathbf{x})\| \leq 2\epsilon K \exp(\beta E(\mathbf{x})) = 2C_1 K \exp(\beta E(\mathbf{x}))\sqrt{\frac{D}{Y}}, \quad (47)$$

giving us the right-hand side of eq. (33). $\qquad\square$

*Proof of theorem* 2. Expanding out the divergence $D^{(L)}$ after $L$ energy descent steps, and using the fact that $\mathbf{x}^{(0)} = \hat{\mathbf{x}}^{(0)} = \mathbf{x}$, we have

$$D^{(L)} \triangleq \|\mathbf{x}^{(L)} - \hat{\mathbf{x}}^{(L)}\| \quad (48)$$

$$= \left\|\left(\mathbf{x}^{(0)} - \sum_{t\in[\![L]\!]}\eta\nabla_{\mathbf{x}}E(\mathbf{x}^{(t-1)})\right) - \left(\hat{\mathbf{x}}^{(0)} - \sum_{t\in[\![L]\!]}\eta\nabla_{\mathbf{x}}\hat{E}(\hat{\mathbf{x}}^{(t-1)})\right)\right\| \quad (49)$$

$$= \left\|\sum_{t\in[\![L]\!]} -\eta\left(\nabla_{\mathbf{x}}E(\mathbf{x}^{(t-1)}) - \nabla_{\mathbf{x}}\hat{E}(\hat{\mathbf{x}}^{(t-1)})\right)\right\| \quad (50)$$

$$= \left\|\sum_{t\in[\![L]\!]} \eta\left(\nabla_{\mathbf{x}}E(\mathbf{x}^{(t-1)}) - \nabla_{\mathbf{x}}\hat{E}(\hat{\mathbf{x}}^{(t-1)})\right)\right\| \quad (51)$$

$$\leq \sum_{t\in[\![L]\!]} \left\|\eta\left(\nabla_{\mathbf{x}}E(\mathbf{x}^{(t-1)}) - \nabla_{\mathbf{x}}\hat{E}(\hat{\mathbf{x}}^{(t-1)})\right)\right\|. \quad (52)$$

Let us denote the individual terms above as $d^{(t)}$ with $D^{(L)} = \sum_{t\in[\![L]\!]} d^{(t)}$. Also, let us denote by $A = 2C_1 K e^{\beta E(\mathbf{x})}\sqrt{D/Y}$ and by $B = (1 + 2K\beta e^{\beta/2})$. Then writing out the $t$-th term

$$d^{(t)} = \left\|\eta\left(\nabla_{\mathbf{x}}E(\mathbf{x}^{(t-1)}) - \nabla_{\mathbf{x}}\hat{E}(\hat{\mathbf{x}}^{(t-1)})\right)\right\| \quad (53)$$

$$\leq \left\|\eta\left(\nabla_{\mathbf{x}}E(\mathbf{x}^{(t-1)}) - \nabla_{\mathbf{x}}E(\hat{\mathbf{x}}^{(t-1)})\right)\right\| + \left\|\eta\left(\nabla_{\mathbf{x}}\hat{E}(\hat{\mathbf{x}}^{(t-1)}) - \nabla_{\mathbf{x}}\hat{E}(\hat{\mathbf{x}}^{(t-1)})\right)\right\| \quad (54)$$

$$\leq \eta B\|\mathbf{x}^{(t-1)} - \hat{\mathbf{x}}^{(t-1)}\| + \eta A, \quad (55)$$

where the first term is bounded using lemma 2 and the definition of $B$, and the second term is bounded using lemma 3 and the definition of $A$.

Note that this gives us the recursion $d^{(t)} \leq \eta A + \eta B D^{(t-1)}$, and thus, $D^{(L)} \leq \sum_{t\in[\![L]\!]} \eta(A + BD^{(t-1)})$.

Writing out the recursion using induction, we can show that

$$D^{(L)}$$

$$= \eta A \left( \sum_{t \in [\![L]\!]} 1 + \sum_{t \in [\![L]\!]} t(\eta B) + \sum_{t \in [\![L]\!]} \sum_{t_1 \in [\![L]\!]} t_1 (\eta B)^2 + \sum_{t \in [\![L]\!]} \sum_{t_1 \in [\![L]\!]} \sum_{t_2 \in [\![t_1]\!]} t_2 (\eta B)^3 \right.$$

$$\left. + \cdots + \sum_{t \in [\![L]\!]} \sum_{t_1 \in [\![L]\!]} \cdots \sum_{t_{L-1} \in [\![t_{L-2}]\!]} t_{L-1} (\eta B)^{L-1} \right) \tag{56}$$

$$\leq \eta A \left( L + L(\eta B L) + L(\eta B L)^2 + \cdots + L(\eta B L)^{L-1} \right) \tag{57}$$

$$= \eta A L \frac{1 - (\eta B L)^L}{1 - \eta B L}. \tag{58}$$

Replacing the values of $A$ and $B$ above gives us the statement of the theorem. $\qquad\square$

### F.2.2 Dependence on the initial energy $E(\mathbf{x})$ of the input.

The divergence upper bound in eq. (12) (in theorem 2) depends on the term $\exp(\beta E(\mathbf{x}))$. However, note that, for the energy function defined in eq. (11), assuming that all memories and the initial queries are in a ball of diameter 1 (which is the assumption A1 in theorem 2), $E(\mathbf{x}) \leq \frac{1}{2} - \frac{\log K}{\beta}$, implying that $\exp(\beta E(\mathbf{x})) \leq \exp(\beta/2)/K$, and we can replace this in the upper bound and remove the dependence on $E(\mathbf{x})$.

However, an important aspect of our analysis is that the bound is input specific, and depends on the initial energy $E(\mathbf{x})$. As discussed above, this can be upper bounded uniformly, but our bound is more adaptive to the input $\mathbf{x}$.

For example, if the input is initialized near one of the memories, while being sufficiently far from the remaining $(K-1)$ memories, then $\exp(\beta E(\mathbf{x}))$ term can be relatively small. More precisely, with all memories and queries lying in a ball of diameter 1, let the query be at a distance $r < 1$ to its closest memories, and as far as possible from the remaining $(K-1)$ memories. In this case, the initial energy $E(\mathbf{x}) \approx -(1/\beta)\log(\exp(-\beta r/2) + (K-1)\exp(-\beta/2))$, implying that

$$\exp(\beta E(\mathbf{x})) \approx \frac{\exp(\beta r/2)}{(1 + \exp(-\beta(1-r)/2))} \leq \exp(\beta r/2). \tag{59}$$

For sufficiently small $r < 1$, the above quantity can be relatively small. If, for example, $r \sim O(\log K)$, then $\exp(\beta E(\mathbf{x})) \sim O(K^\beta)$, while $r \to 0$ gives us $\exp(\beta E(\mathbf{x})) \to O(1)$. This highlights the adaptive input-dependent nature of our analysis.

