# OpenReview forum: "Dense Associative Memory Through the Lens of Random Features"
_NeurIPS.cc/2024/Conference — NeurIPS 2024 poster_

### Official Review · Reviewer_9NBh · 2024-06-12

**Soundness:** 2
**Presentation:** 3
**Contribution:** 3
**Rating:** 7
**Confidence:** 4

**Summary:**

This paper introduce an iterative random feature Dense Associative Memory Model (DenseAM).
It is a kernel approximation of the standard DenseAMs using random feature decomposition and approximation.
The authors term this approach to DenseAMs as distributed representation/formulation of memory (DrDAM).

Theoretically, they analyze the time complexity of the proposed method (Thm1) and the approximation error (Thm2), both benchmarked against the standard DenseAM (MrDAM).
Experimentally, they evaluate the approximation accuracy of DrDAM's energies and gradients, defined as the MAE difference between DrDAM and MrDAM, and the retrieval accuracy of DrDAM.
Importantly, their analysis of DrDAM considers its iterative nature, distinguishing their contribution from prior works.

In sum, they address important research questions and their approach is natural and intuitive based on recent advances in the random feature kernel trick.

**Strengths:**

* **Originality**: This paper expands our understanding of Dense Associative Memory Models (DenseAMs) by presenting a random feature DenseAM, termed DrDAM. The idea of DrDAM is novel and beneficial to the ML and associative memory community.

* **Significance**: While the idea of random feature DenseAM is not entirely new, this work delves deeper than most prior studies. For example, to the best of my knowledge, there is nothing as concrete as the iterative Algorithm 1 presented in prior works. This represents a significant and solid step forward for the community, and I appreciate the effort.

* **Theoretical Contribution**: This paper provides an efficiency and approximation analysis of DrDAM.

* **Experimental Contribution**: They explore some aspects of DrDAM, including the approximation of energies and gradients and the approximation ability of DrDAM. Reproducible code is also provided.

**Weaknesses:**

## Summary of Weaknesses

Overall, this paper presents a good idea, but it seems rushed, which affects the contribution of an otherwise very interesting work. Regardless of the final decision, I hope the authors find my comments helpful in further refining their draft.

There are a few areas that could benefit from further improvement and clarity:

* **Experiments:** The current experimental results would be strengthened by including comparisons with established baselines and more ablation studies.
* **Clarity:** Some of the theoretical results are difficult to interpret due to minor typos and a need for increased mathematical clarity.
* **Motivation:** The motivation and problem statement could be articulated more convincingly. Even after multiple readings of the intro, it remains unclear if distributed representation is truly parameter-efficient beyond the nonparametric sense.
* **Related Works:** Additionally, while not critical, the idea of Random Feature DenseAM (and Kernel DenseAM) has been discussed in prior works. The related discussion in this paper could be expanded to better position the contribution within the existing literature. For examples:
  * ```Random Feature DenseAM``` Random Feature Hopfield Networks generalize retrieval to previously unseen examples https://openreview.net/forum?id=bv2szxARh2)
  * ```Random Feature DenseAM``` Nonparametric Modern Hopfield Models https://arxiv.org/abs/2404.03900
  * ```Kernel DenseAM``` Kernel Memory Networks: A Unifying Framework for Memory Modeling  https://arxiv.org/abs/2208.09416
  * ```Kernel DenseAM``` Uniform Memory Retrieval with Larger Capacity for Modern Hopfield Models  https://arxiv.org/abs/2404.03827
  * ```Kernel DenseAM``` Bridging Associative Memory and Probabilistic Modeling https://arxiv.org/abs/2402.10202



**Note:** I have tried my best to evaluate this paper. I spent much more time on this review than on my previous ones because I like this paper. Still, I might have missed something. Please correct me if I am wrong with fair justifications. I am open to changing my opinion and raising the score.

---

## Major: My main concern with this paper is its soundness and inadequate experimental evidence

### General


* ```line23:``` "Memory representation" is a strange term for me. If you are referring to the way of encoding memory patterns onto the energy landscape, isn't it "memory encoding"? Representation generally involves learning, and to my understanding, there is no learning here.

  >Thus, in situations when the set of the memories needs to be expanded by introducing new patterns one must introduce additional weights.

  A side comment: this statement implies nonparametric, as suggested in [17].

* ```distributed representation```: I find the description of distributed representation vague. With more memories, $K$ increases, resulting in more degrees of freedom in $\mathbf{T}$. You can say the total number of entries in $\mathbf{T}$ is fixed, but not the number of parameters, unless you clearly define "parameters" differently. The current draft makes it difficult to parse the soundness.  I believe the idea of distributed representation deserves more elaboration. The current draft seems nonparametric to me.

  In my current review, I assume what is stated in lines 29-30 is correct.

* ```Sec3```: Section 3 is hard to follow due to typos and the lack of clarity.

* ```Missing Broader Impacts```: The authors didn't discuss both potential positive and negative societal impacts except in the checklist. My understanding is that this discussion needs to be included in the main text or the appendix, as the checklist is not considered part of the paper.

---

### Theory


* ```line92```: Eqn. (6) is hard to follow without clearly defined $S,s$.

  I suppose $S:\mathbb{R}^d\times \mathbb{R}^d\to \mathbb{R}$ which makes $s\in\mathbb{R}$ a scalar realization of $S$?


* ```No fixed point convergence results```: Per my understanding, DenseAMs are energy models with memories stored in their local minima. Memory retrieval is conducted through energy minimization. This setting requires convergence between local minima and fixed points of retrieval updates, such as the convergence results of [Ramsauer2020,Hu2023]. Otherwise, there may be unbounded retrieval error due to the mismatch between local minima and fixed points of retrieval updates. Specifically, I believe lines 106-110 are at best ambiguous.

  * [Ramsauer2020] Hopfield Networks is All You Need. Ramsauer, H., Schäfl, B., Lehner, J., Seidl, P., Widrich, M., Adler, T., Gruber, L., Holzleitner, M., Pavlović, M., Sandve, G.K., and Greiff, V., 2020. arXiv preprint arXiv:2008.02217.

  * [Hu2023] On Sparse Modern Hopfield Model. Hu, J.Y.C., Yang, D., Wu, D., Xu, C., Chen, B.Y., and Liu, H., 2023. Advances in Neural Information Processing Systems, 36.

* ```Theorem 1```: I feel Thm1 may be incorrect. Could you elaborate more on this?

  * **My understanding:** Let $L$ be the number of updates, $K$ be the number of stored memories, $D$ be the memory dimension, and $Y$ be the feature space dimension. DrDAM improves time complexity from $O(LKD)$ to $O(D(Y K + L(Y + D))$, and improves peak memory from $O(KD)$ to $O(Y+D)$.
  * **Comment1:** $O(D(Y K + L(Y + D))$ is clearly a typo and makes the result hard to parse.
  * **Comment2:** Without specifying the relationships between $L$, $K$, $D$, and $Y$, it's hard to tell if this is truly an improvement. I can easily think of counterexamples.

  Given these, can you kindly elaborate more on the significance of thm1?


* ```Theorem 2```: Eqn. (12) is strange. Is there any guarantee ensuring $\alpha L(1+2K\beta e^{\beta/2}) < 1$? If not, it doesn't make sense that more updates would result in larger error. Perhaps this is related to the missing fixed point convergence?

---

### Experiments

* ```No Comparison to existing works:``` there is no baseline compared.

* ```No efficiency evaluation1:``` Please include ablation studies on changing $L,K,D,Y$ to verify Thm1.

* ```No efficiency evaluation2:``` Please also compare the efficiency with other methods, including linear, top-K, random feature, and random masked from [17], as well as dense [Ramsauer2020], sparse [Hu2023], and generalized sparse [Wu2023] MHNs. For example, the current draft needs some figures similar to Fig. 4 of [17] to justify the efficiency. Please do whatever makes sense to you. For instance, you can pick just two out of [Ramsauer2020, Hu2023, Wu2023] since they share similar structures.

  * [Ramsauer2020] Hopfield Networks is All You Need. Ramsauer, H., Schäfl, B., Lehner, J., Seidl, P., Widrich, M., Adler, T., Gruber, L., Holzleitner, M., Pavlović, M., Sandve, G.K., and Greiff, V., 2020. arXiv preprint arXiv:2008.02217.

  * [Millidge2022] Universal Hopfield Networks: A General Framework for Single-Shot Associative Memory Models. Millidge, B., Salvatori, T., Song, Y., Lukasiewicz, T., and Bogacz, R., 2022, June. In International Conference on Machine Learning (pp. 15561-15583). PMLR.

  * [Wu2023] Stanhop: Sparse Tandem Hopfield Model for Memory-Enhanced Time Series Prediction. Wu, D., Hu, J.Y.C., Li, W., Chen, B.Y., and Liu, H., 2023. arXiv preprint arXiv:2312.17346.

  * [Hu2023] On Sparse Modern Hopfield Model. Hu, J.Y.C., Yang, D., Wu, D., Xu, C., Chen, B.Y., and Liu, H., 2023. Advances in Neural Information Processing Systems, 36.



* ```Compare with existing kernel DenseAMs.``` Also, a direct comparison with existing kernel methods is also necessary. For example,
  * Uniform Memory Retrieval with Larger Capacity for Modern Hopfield Models. Wu, D., Hu, J.Y.C., Hsiao, T.Y., and Liu, H., 2024. arXiv preprint arXiv:2404.03827.

  * Kernel Memory Networks: A Unifying Framework for Memory Modeling. Iatropoulos, G., Brea, J., & Gerstner, W., 2022. Advances in Neural Information Processing Systems, 35, 35326-35338.

  * Bridging Associative Memory and Probabilistic Modeling. Schaeffer, R., Zahedi, N., Khona, M., Pai, D., Truong, S., Du, Y., Ostrow, M., Chandra, S., Carranza, A., Fiete, I.R., and Gromov, A., 2024. arXiv preprint arXiv:2402.10202.

---

## Minor

* ```Inadequate Related Work Discussion```: A paragraph discussing related works is given at the end of the introduction. However, I feel it could be made more comprehensive to provide more background on the evolution of ideas and existing works in this field. At least, I feel the current draft is hard for a non-expert in DenseAMs to follow. Please see the following points.

* ```line38:``` I believe here can be benefited by including more comprehensive references. There are other recent works showing exponentially large memory storage capacity in various settings and dense associative memory (or modern Hopfield) networks/models. For examples: (I am not sure of their exact relevance, so I will leave it to the authors to decide whether they should be included and discussed.)

   * Hopfield Networks is All You Need. Ramsauer, H., Schäfl, B., Lehner, J., Seidl, P., Widrich, M., Adler, T., Gruber, L., Holzleitner, M., Pavlović, M., Sandve, G.K., and Greiff, V., 2020. arXiv preprint arXiv:2008.02217.
  * Kernel Memory Networks: A Unifying Framework for Memory Modeling. Iatropoulos, G., Brea, J., & Gerstner, W., 2022. Advances in Neural Information Processing Systems, 35, 35326-35338.
  * On Sparse Modern Hopfield Model. Hu, J.Y.C., Yang, D., Wu, D., Xu, C., Chen, B.Y., and Liu, H., 2023. Advances in Neural Information Processing Systems, 36.
  * Stanhop: Sparse Tandem Hopfield Model for Memory-Enhanced Time Series Prediction. Wu, D., Hu, J.Y.C., Li, W., Chen, B.Y., and Liu, H., 2023. arXiv preprint arXiv:2312.17346.
  * Bridging Associative Memory and Probabilistic Modeling. Schaeffer, R., Zahedi, N., Khona, M., Pai, D., Truong, S., Du, Y., Ostrow, M., Chandra, S., Carranza, A., Fiete, I.R., and Gromov, A., 2024. arXiv preprint arXiv:2402.10202.
  * Sparse and Structured Hopfield Networks. Santos, S., Niculae, V., McNamee, D., and Martins, A.F., 2024. arXiv preprint arXiv:2402.13725.
  * On Computational Limits of Modern Hopfield Models: A Fine-Grained Complexity Analysis. Hu, J.Y.C., Lin, T., Song, Z., and Liu, H., 2024. arXiv preprint arXiv:2402.04520.
  * Outlier-Efficient Hopfield Layers for Large Transformer-Based Models. Hu, J.Y.C., Chang, P.H., Luo, R., Chen, H.Y., Li, W., Wang, W.P., and Liu, H., 2024. arXiv preprint arXiv:2404.03828.


* ```line46:``` Similarly, here can be benefited by including more recent references
  > DenseAM family [...]

* ```line51```: A similar motivating question is asked and explored in [17] from a kernel regression perspective. I agree this paper contributes something different and even beyond [17]. However, given the similarity of aims between the two, the current draft should:
  1. State the overlaps between the two works (both use a kernel approach to DenseAMs, and...).
  2. State the clear distinctions between the two works (both use a kernel approach to DenseAMs, but...).

  Such discussion will help readers parse the contributions of this work, which is currently lacking.

  [17] Jerry Yao-Chieh Hu, Bo-Yu Chen, Dennis Wu, Feng Ruan, and Han Liu. Nonparametric modern hopfield models. arXiv preprint arXiv:2404.03900, 2024. URL https://arxiv.org/pdf/2404.03900.pdf.


* ```line92```:
  > those results have been recently applied to associative memory [15].

  I might have missed something, but it would be better to make the discussion of prior work more complete if it's presented in its current form (e.g., a paragraph highlighting the novelty and contributions benchmarked against existing works). The application of the kernel trick to associative memory is not new. While [16,17] are discussed right after [15], I feel it's better to cite them together as [15,16,17]. Additionally, there are two papers related to the kernel trick in associative memory, which I have attached below. Again, I am not sure of their exact relevance, so I will leave it to the authors to decide whether they should be included and discussed.

  - Iatropoulos, G., Brea, J., & Gerstner, W. (2022). Kernel memory networks: A unifying framework for memory modeling. Advances in neural information processing systems, 35, 35326-35338.

  -  Schaeffer, R., Zahedi, N., Khona, M., Pai, D., Truong, S., Du, Y., Ostrow, M., Chandra, S., Carranza, A., Fiete, I.R. and Gromov, A., 2024. Bridging Associative Memory and Probabilistic Modeling. arXiv preprint arXiv:2402.10202.

* ```line97```:
  > Another paper [17] uses kernels for studying the sparse modern Hopfield network.

  This is incorrect.

  As stated in their abstract, [17] presents a general framework (referred to as nonparametric) to analyze and design modern Hopfield networks. The sparse network is just a special case of their framework. Additionally, [17] has already introduced the construction of random feature HMNs. While this construction lacks detailed analysis, which is a limitation they acknowledged, it would be more accurate to say that the submitted draft serves as a strong companion to [17] with rigorous theoretical and empirical results. I understand that the authors are not obliged to treat arXiv preprints (unpublished papers) too seriously, but it is better to give accurate credit to existing works when possible. To clarify, I am very happy to see this draft improves the results/proposal of [17] with detailed analysis, including multiple retrieval updates.

* I did not check the proofs line by line. However, I had a quick skim through them and found some typos and areas for improvement. Please do another round of proofreading if possible.


---

### Update 2024/08/07: raise score from 4 -> 5 -> 6 -> 7

**Questions:**

* ```line109```: why you call the number of updates as *layers*? Are you recurrently passing the output as input to the same network or the network really has many layers? For example, recurrent layer is RNN sense is the later.


* ```Proposition 1,2,3```: what is peak memory? why we should care about it?

* ```Time Complexity```: I appreciate the analysis of the computational complexity of the proposed method. From my view, this proposal is a special case of [Hu2024] with $L=1$ (where $L$ is the sequence length in their analysis). [Hu2024] provides a characterization of all sub-quadratic (efficient) variants of the modern Hopfield model. Essentially, your proposal is an efficient approximation to the softmax DenseAMs. Hence, the relevance holds. Can you discuss how your proposal fits into the results of [Hu2024]?

  * [Hu2024] Hu, J.Y.C., Lin, T., Song, Z. and Liu, H., 2024. On Computational Limits of Modern Hopfield Models: A Fine-Grained Complexity Analysis. arXiv preprint arXiv:2402.04520.


* ```line255, Exp Observation 2```: This observation is a bit counterintuitive. Normally, large $\beta$ means low-temperature region and leads to more stable/accurate retrieval, e.g., infinity $\beta$ leads to argmax-retrieval. Why random DrDAM shows otherwise?



* ```Fig3```: Intuitively, why larger $\beta$ needs larger $Y$?

* ```line303```: What's the point of approximating energies and update dynamics? Shouldn't the significance lie in the approximation error of retrieval?

* I am wondering if you have explored the learning aspect of the proposed method. It is known that many DenseAMs are connected to transformer attention and can be used as a learning model or layer. Can you explore this part a bit as in [17]? If not, please explain why.

* Can you kindly remind me which part of this work supports the claim "making it possible to introduce new memories without increasing the number of weights"?

**Limitations:**

There is a limitation section but no broader impact or impact statement expect in the checklist.


Appendix A of the draft acknowledges that the experimental explorations are limited. However, it would benefit from discussing comparisons with baselines and including some vital ablations. This would help clarify the contribution and soundness of the proposed method. See above for other limitations.

---

> ### Author Rebuttal · Authors · 2024-08-06
>
> We thank the reviewer for the thorough and insightful evaluation of our work. We are glad you found it interesting and beneficial to the ML community!
>
> We will add a Related Work section in which we will review the results of
>
> - Hu et al., Nonparametric Modern Hopfield Models
> - Wu et al., Uniform Memory Retrieval
>
> which are already referenced in our paper, in addition to
>
> - Negri et al., Random Feature Hopfield Networks generalize…
> - Hu et al., On Sparse Modern Hopfield Model
> - Wu et al., Stanhop
> - Iatropoulos et al., Kernel Memory Networks
>
> and other suggested papers. We will also include a Broader Impact statement. Below, we do our best to address your comments within the character constraints of the rebuttal.
>
> > …results would be strengthened by including comparisons with established baselines and more ablation studies.
> >
>
> We conducted numerical experiments to validate our theorems (see 1-page pdf). Note that our main goal is to approximate the energy and trajectory of a standard DenseAM (MrDAM) using Random Features. Thus, MrDAM is the “baseline” for our method.
>
> > "Memory representation" is a strange term for me…
> >
>
> We use the word representation in a more colloquial sense - the description in terms of something. Similarly to Fourier representation, momentum representation in physics, etc.
>
> > You can say the total number of entries in $T$ is fixed, but not the number of parameters, unless you clearly define "parameters" differently…
> >
>
> We define the number of parameters as the number of entries in $T$. We will include additional elaboration on these aspects in the revised paper.
>
> > I feel Thm1 may be incorrect
> >
>
> (Re: Comment 1) We apologize for the typo. The bound should be $O(D(YK + L(Y+D)))$; the version in the paper is missing a closing bracket around $O(\cdot)$. Other than that, the bound is as stated, with the $DYK$ term coming from the `ProcMems` subroutine and the $LD(Y+D)$ term coming from the $L$ `GradComp` invocations. See the proof of Thm 1 in App E1.
>
> (Re: Comment 2) Thm 1 quantifies the complexity of procedures in Alg 1. We do not claim improvement over standard MrDAM for the precise reason stated by the reviewer regarding the choices of $D, Y, K, L$.
>
> These results highlight that energy descent with DrDAM requires computation and memory that only depends on $D$ and $Y$. Together with Thm 2 and Cor 1, we characterize where the energy descent divergence between MrDAM and DrDAM can be bounded with a choice of $Y$ that only depends on $D$ (and other parameters in the energy) but not $K$.
>
> > Is there any guarantee ensuring $\alpha L (1 + 2K \beta e^{\beta/2}) < 1$?
> >
>
> There is no guarantee, as stated in Thm 2. Denoting this term with $\delta = \alpha L(1 + 2K \beta e^{\beta/2})$, the upper bound contains a term of the form $\left(\frac{1 - \delta^L}{1 - \delta}\right)$. If $\delta < 1$, it is clear that the numerator grows with $L$, providing a worse upper bound. However, if $\delta > 1$, $\frac{1 - \delta^L}{1 - \delta}$ can be equivalently written as $\frac{\delta^L - 1}{\delta - 1}$, where the $\delta^L$ term grows with $L$, still having the same effect of larger divergence upper bound with larger $L$.
>
> > **Re: proof for “fixed point convergence results”**
> >
>
> The proof is standard in DenseAM literature (see appendices in [paper1](https://arxiv.org/abs/2008.06996) or [paper2](https://arxiv.org/abs/2107.06446)). Consider our Eq (11) for the energy as an example, whose dynamics is described by the following eqs ($i=1...D$)
>
> $$
> \tau \frac{dx_i}{dt} = - \frac{\partial E}{\partial x_i}.
> $$
>
> For this reason, the energy monotonically decreases on the dynamical trajectory
>
> $$
> \frac{dE}{dt} = \sum\limits_{i=1}^D \frac{\partial E}{\partial x_i} \frac{dx_i}{dt} = -\tau \sum\limits_{i=1}^D \Big( \frac{dx_i}{dt}\Big)^2\leq 0.
> $$
>
> The energy in Eq (11) is bounded from below, by $-\log(K)/\beta$, since the argument of the logarithm is bounded from above by $K$. Thus the energy descent dynamics has to stop sooner or later when $\frac{dx_i}{dt}=0$, which defines the fixed points. Thus, local minima of the energy correspond to the fixed points of the dynamics.
>
> > …include ablation studies on changing $L,K,D,Y$ to verify Thm1.
> >
>
> This is a great suggestion. However, note that Thm 1 is a tight analysis of Alg 1, counting the $D$- and $Y$-dimensional vector initializations, additions, and dot-products. So we did not think such an ablation study would provide additional insight.
>
> > Why larger $\beta$ needs larger $Y$?
> >
>
> The reviewer is correct that large $\beta$ (low-temperature) leads to more stable and accurate retrieval  by approaching the argmax-retrieval in standard DenseAMs. However, this regime is specifically difficult for random features. Consider the L2 distance based energy in Eq (11):
>
> $$
> E(\mathbf{x}) = -\frac{1}{\beta} \log \sum_{\mu} \exp\left(-\frac{\beta}{2} \| \boldsymbol{\xi}_\mu - \mathbf{x} \|^2 \right).
> $$
>
> As $\beta$ increases, the $\exp(\cdot)$ term, which we approximate with random features, gets skewed to the extremes across the $K$ memories, with the $\exp(\cdot) \to 0$ for all but the closest memory. This makes the RF approximation harder, requiring larger $Y$ for better approximation.
>
> > which part of this work supports the claim that "making it possible to introduce new memories without increasing the number of weights"?
> >
>
> Prop. 5 shows how to update weights $\mathbf{T}$ with a new memory $\boldsymbol{\xi}$ without increasing the number of weights:
>
> $$
> \mathbf{T} \gets \mathbf{T} + RF(\tau, \boldsymbol{\xi}),
> $$
>
> taking $O(DY)$ time and $O(D+Y)$ memory.

---

> > ### Comment · Reviewer_9NBh · 2024-08-07
> >
> > Thanks for clarifications. Haven't finished going through them. Yet, I understand the rebuttal space is limited. I am ok if the authors use comments to clarify more. I will read them.

---

> > ### Comment · Reviewer_9NBh · 2024-08-07
> >
> > > We conducted numerical experiments to validate our theorems (see 1-page pdf). Note that our main goal is to approximate the energy and trajectory of a standard DenseAM (MrDAM) using Random Features. Thus, MrDAM is the “baseline” for our method.
> >
> > Thanks for the clarification.
> >
> > Since it's approximation to single step of iterative algorithm, can you comment on the "iterative" error bound? Is it sth that you analyze or the current results are just for single step?
> >
> > Thanks!
> >
> > > We use the word representation in a more colloquial sense - the description in terms of something. Similarly to Fourier representation, momentum representation in physics, etc.
> >
> > This is ok. Yet I suggest the authors to refine the wording to ensure preciseness. The examples you mentioned, Fourier & momentum, are mathematically meaningful, e.g., one of the domains connected by some transformation.
> >
> > > (Re: Comment 2) Thm 1 quantifies the complexity of procedures in Alg 1. We do not claim improvement over standard MrDAM for the precise reason stated by the reviewer regarding the choices of $L,Y,D,K$
> >
> > Thanks for clarification. Yet this confuses me. I suppose random feature is for efficiency by sacrificing some accuracy, but you don't really ensure DrDAM is more efficient here. Isn't this make the contribution vacuous?
> >
> > > There is no guarantee, as stated in Thm 2...
> >
> > This is concerning. The multiple updates of DrDAM lead to looser error bound. This contradicts to the fixed point convergence argument. It is not contraction map toward fixed point.
> >
> > > This is a great suggestion. However, note that Thm 1 is a tight analysis of Alg 1, counting the .. and ...-dimensional vector initializations, additions, and dot-products. So we did not think such an ablation study would provide additional insight.
> >
> > I am confused. What do you mean by tight analysis? Do you mean "everything is exact by counting"? Thm 1 is about # of operations needed, correct?
> >
> > ===
> >
> > Thanks for your response. Please address above if possible. Thank you!

---

> > > ### Author Response · Authors · 2024-08-07
> > >
> > > > Since it's approximation to single step of iterative algorithm, can you comment on the "iterative" error bound? Is it sth that you analyze or the current results are just for single step?
> > > >
> > >
> > > Nothing in our work is about a single step update. We only consider fully recurrent DenseAMs with many iterative updates. Just to reiterate, the network is defined by the differential equation
> > >
> > > $$
> > > \tau \frac{dx_i}{dt} = -\frac{\partial E}{\partial x_i},
> > > $$
> > >
> > > which can be discretized to obtain Eq (5) in the submission. All the results in the paper pertain to studying this fully iterative system. The number of iterations can be as big as it is necessary in order to reach the fixed point.
> > >
> > > The fixed points of this trajectory are the "memories" of the system. MrDAM will follow this trajectory by descending the gradient of the standard energy $-\frac{\partial E}{\partial x_i}$ in e.g., Eq (11). DrDAM will follow this trajectory by descending the gradient of the approximate energy $-\frac{\partial \hat{E}}{\partial x_i}$.
> > >
> > > To be a "good" approximation, DrDAM must produce similar energies and fixed points as that of MrDAM (Fig 2 and 3). We do not specifically consider the case of single steps down the energy landscape for either MrDAM or DrDAM because we want to guarantee that we have reached the "end" (fixed point) of the dynamics. See the trace of energy over time in Fig 1 to see that a single update step is not guaranteed to converge to a fixed point.
> > >
> > > This said, if you are interested in how well DrDAM approximates MrDAM in the single step regime, you can substitute $L=1$ Thm 2, taking 1 discrete step down the energy. The approximation error in this regime is dependent only on the error between the gradients $\nabla \hat{E}$ and $\nabla E$, which we empirically test in Fig 2 of our paper.
> > >
> > > > I suppose random feature is for efficiency by sacrificing some accuracy.
> > > >
> > >
> > > We use Random Features to decouple the energy and dynamics of MrDAM from the number of stored patterns $K$. Efficiency is not our goal in this paper. Arguments for the "efficiency" of DrDAM over MrDAM depend on specific choices for $L,Y,D,K$. Thm 1 derives the computational complexity as a function of these hyperparams; Thm 2 derives the error bounds as a function of these hyperparams. In addition, Fig 2 and 3 empirically characterize where DrDAM is a good approximation i.e., when queries have low initial energy at or near the stored patterns, at higher values of $Y$, and at lower values of $K$ and $\beta$. Thus, a user can use the relationships described in our work to choose regimes where DrDAM is more or less efficient than MrDAM. This was our intent when we said "We do not claim improvement over standard MrDAM for the precise reason stated by the reviewer regarding the choices of $L,Y,D,K$”.
> > >
> > > > What do you mean by tight analysis?... Thm 1 is about # of operations needed, correct?
> > > >
> > >
> > > Correct, Thm 1 is about the number of operations needed to perform memory retrieval using DrDAM. The complexity can be verified by counting the number of operations during retrieval. Thm 1 emphasizes that the gradient computation `GradComp` is independent of the value of $K$.
> > >
> > > > The multiple updates of DrDAM lead to looser error bound. This contradicts to the fixed point convergence argument.
> > > >
> > >
> > > DenseAM in continuous time is a fully contracting system, if the energy decreases along the dynamical trajectory and the energy is bounded from below (please see our rebuttal for the derivation of this result). Both these conditions are satisfied for both MrDAM and DrDAM, thus both these networks are contracting. The bound derived in Eq 12 is for discretized system, Eq (5). You are correct that this bound is useless for $\delta>1$, as it grows to infinity as $L$ is increased. However, it is a very useful bound for sufficiently small $\alpha$, which corresponds to $\delta<1$ (please see Corollary 1). There is no contradiction here, it’s just the bound becomes uninformative for $\delta>1$. As a side note, in our empirical experiments $\alpha$ is always sufficiently small to make sure that $\delta<1$ and that the discretized network is sufficiently close to the network described by continuous time.

---

> ### Comment · Reviewer_9NBh · 2024-08-08
>
> Thanks for the clarifications.
>
> 1. Maybe the authors should polish the draft regarding Thm1, specifying under which conditions it is efficient, as efficiency is mentioned in the draft. I still feel that it is necessary to have exps showing these efficient conditions are true in the final version.  Please correct me if I am wrong. 2 settings (efficient v.s. inefficient) benchmarked against MrDAM should suffice.
>
> 2. I'd like to remind the authors that the convergence of fixed point and stable point of the energy landscape (local minima) are different concepts. It seems that your derivation is about the latter, given that it's based on gradient descent on $E$. I understand the intuition you want to convey, but I remind the authors that this physics style of derivation is not really fixed-point convergence in a mathematically rigorous way. For one, you don't provide a convergence rate; you just assume it converges perfectly at $\nabla_x E = \frac{dx}{dt} = 0$, whereas, in reality, you only converge to a region defined by the termination of the gradient descent algorithm. Moreover, can you clarify if $E$ is convex? If it is, such analysis should be easier to include.
>
> 3. If your derivation is about stable point of energy landscape (local minima), then how can you ensure fixed point convergence? It is acceptable that you don't have it or don't know how to prove it in this model (leave for future work), but it's important to be precise and accurate.
>
> 4. My main concern is that it's impossible to have $\delta <1$ given $\alpha,L,K$ are all positive constants greater than 1.
> It's strange to have an **iterative** approximation algorithm with error exponentially scaling with its iteration number. This makes Thm2 vacuous. Please correct me if I am wrong.
>
> I appreciate the authors' efforts making this work clearer to me so far. I learned a lot. For this, I raise my score from 4 to 5.
>
> I am willing to further raise score if above concerns are addressed: exps for efficient criteria, fixed point convergence and Thm2.
>
> Thank you!

---

> > ### Author Response · Authors · 2024-08-09
> >
> > Thank you for your questions. We are happy to provide further clarity.
> >
> > > Maybe the authors should polish the draft regarding Thm1, specifying under which conditions it is efficient, as efficiency is mentioned in the draft. I still feel that it is necessary to have exps showing these efficient conditions are true in the final version.
> > >
> >
> > We understand that the reviewer is asking for us to choose configurations of the hyperparams in Thm 1 that cause DrDAM to be more/less efficient (in terms of memory and computational complexity) than the baseline MrDAM. Per your repeated requests, we will add experiments on select configurations to the final draft, though “efficiency” was not an emphasis of our original draft. On re-reading our submission, we find mention to efficiency in only one sentence, in our Conclusion: “We have demonstrated how this can be done efficiently…” We will modify this sentence to prevent future confusion. Thank you for identifying this. Just to reiterate, theoretically, the peak memory complexity of MrDAM is $O(KD)$ vs $O(Y+D)$ for DrDAM. Given that in the interesting regime $Y\sim D/\varepsilon^2$ (Corollary 1), DrDAM is more memory efficient than MrDAM if $K>const/\varepsilon^2$. For running time complexity (ignoring memory encoding), which is $O(LKD)$ for MrDAM vs. $O(LD(Y+D))$ for DrDAM, MrDAM is generally more efficient than DrDAM.
> >
> > > …the convergence of fixed point and stable point of the energy landscape (local minima) are different concepts…
> > >
> >
> > Actually, in the DenseAM energies studied in this work, there is a 1:1 correspondence between fixed points and local minima of the energy function. The argument we presented above is mathematically rigorous and correct. For intuition, note that the energy landscape of Eq (11) “looks like” a mixture of Gaussians that has been inverted s.t. local peaks in log-probability are now local minima in energy. Thus, our model is non-convex (i.e., there can be many local minima) and each local minimum is a “fixed point” (where $\nabla_{\mathbf{x}} E = \frac{dx}{dt} = 0$) of the dynamics.
> >
> > Imagine for simplicity that $\beta$ is large and $x_i$ is sufficiently close to one of the memories, say memory 1 (i.e., $\xi^1_i$), then only one of the exponential terms in Eq 11 is appreciably different from 0. Thus, in this limit the energy can be simplified to $E \approx \frac{1}{2} \sum\limits_{i=1}^D (\xi^1_i - x_i)^2$, locally around memory 1. This is just a quadratic function. Optimization is very simple and the solution for the fixed point is $x_i=\xi^1_i$ both from solving the continuous time differential equation and from minimizing the energy. There are no any subtleties or ambiguities here.
> >
> > > can you clarify if $E$ is convex?
> > >
> >
> > The energies of DenseAMs are non-convex functions because they have many local minima.
> >
> > > For one, you don't provide a convergence rate… you only converge to a region defined by the termination of the gradient descent algorithm
> > >
> >
> > A convergence rate is not necessary to prove convergence. By the argument above, the fixed point “regions” where gradient descent terminates are in fact points, which are indeed local minima of the energy function. Maybe you refer to some numerical precision errors? If so, of course everything that we do is impacted by those errors, but they are small. So, in reality we always stop $10^{-6}$ or so away from the true fixed point, but this does not impact any of our methods.
> >
> > > If your derivation is about stable point of energy landscape (local minima), then how can you ensure fixed point convergence?
> > >
> >
> > We believe we have answered this question above.
> >
> > > it's impossible to have $\delta < 1$ given $\alpha, L, K$ are all positive constants greater than 1.
> > >
> >
> > This statement incorrectly assumes that $\alpha > 1$: $\alpha$ (our discrete gradient-descent step size) is a strictly positive scalar that approaches the continuous setting as $\alpha \rightarrow 0$. In all the results presented in our paper $\alpha<1$.

---

> ### Comment · Reviewer_9NBh · 2024-08-09
>
> > Actually, in the DenseAM energies studied in this work, there is a 1:1 correspondence between fixed points and local minima of the energy function...
>
> This is not entirely accurate. Your gradient descent (GD) can't reliably approach the point $\nabla_x E = 0$ given $E$ is nonconvex, at least not without strong assumptions. For example, if your local minima form a connected ring, what would be your fixed point?
>
> There is a line of research dedicated to this subtlety. Please refer to the "Global Convergence Theory of Iterative Algorithms" in [1] for related discussions. In that work, global convergence refers to the convergence of "stable points of $E$" and "fixed points of iterative algorithms." Without proving your fixed-point convergence with similar mathematical rigor, the claim seems overreached.
>
> For the same reason, the following statements are also unconvincing without supporting proofs:
> * > For intuition, note that the energy landscape of Eq (11) “looks like” a mixture of Gaussians that has been inverted such that local peaks in log-probability are now local minima in energy. **Thus**, our model is non-convex (i.e., there can be many local minima) and each local minimum is a “fixed point”...
>
> * > By the argument above, the fixed-point “regions” where gradient descent terminates are, in fact, points, which are indeed local minima of the energy function...
>
> Still, I might be wrong, but for now, I am not convinced by the math presented in this paper.
>
> [1] Sriperumbudur, Bharath K., and Gert RG Lanckriet. "On the Convergence of the Concave-Convex Procedure." NIPS. Vol. 9. 2009.
>
> ---
>
> Regarding $\alpha$, please correct me if I am wrong. In `line 181` of the submitted draft, it states "$\alpha \in \{1,...,Y\}$". Aren't they the same $\alpha$? If I mistook two different $\alpha$, it would be better to change the repeated index notation to avoid confusion.
>
> ---
>
> I appreciate the authors' efforts and trust they will include efficiency experiments (and also modify the draft as they promised). For this, I raise score from 4->5->6.
>
> Thank you!
>
> (Side comment: Even if it's not the main focus of this work, I still think it's important to show general readers where this method is an efficient approximation.)

---

> > ### Comment · Reviewer_9NBh · 2024-08-09
> >
> > **Edited 15mins later:** After further consideration, the fixed-point convergence issue is just food for thought/discussion but not a fatal flaw in this work. The work is still refreshing overall, so I am raising my score to 7. This paper is one of those with real new ideas. I recommend it for acceptance.
> >
> > However, I hope the authors can refine the related discussions based on my comments. Alternatively, if I am mistaken, please clarify where I went wrong. I'd really appreciate it.

---

> > > ### Author Response · Authors · 2024-08-09
> > >
> > > We thank the reviewer for seeing the value in our work and increasing their score! In the spirit of a healthy discussion period, we will continue to respond to your questions:
> > >
> > > > Your gradient descent (GD) can't reliably approach the point $\nabla_x E = 0$ given $E$ is nonconvex, at least not without strong assumptions. For example, if your local minima form a connected ring, what would be your fixed point?
> > > >
> > >
> > > It would require a huge amount of fine-tuning for $\beta$ and the location of memories to engineer a connected ring, or any other type of flat directions in the energy defined by Eq (11). Here we are working in the dense regime of DenseAMs, when $\beta$ is sufficiently large, so that every memory is a point attractor. Flat directions are common in more advanced models of AM, such as [Energy Transformer](https://arxiv.org/abs/2302.07253), but these are not considered in this work.
> > >
> > > > In `line 181` of the submitted draft, it states $\alpha \in \{1,...,Y\}$. Aren't they the same $\alpha$? If I mistook two different $\alpha$, it would be better to change the repeated index notation to avoid confusion.
> > > >
> > >
> > > You are absolutely justified in your confusion and we apologize for this clash of notation, where $\alpha$ can represent both the index into the distributed memory and discrete gradient-descent step size. The $\alpha$ in Eq (12) is the “gradient descent step size” first defined in Eq (5), and used in Cor 1 to constrain the divergence. We will refine this notation in the final draft — thank you for spotting this.

---

> > > > ### Comment · Reviewer_9NBh · 2024-08-09
> > > >
> > > > The connected ring is just a special case that current theory can't justify. This is an example where energy stable points might not coincide with retrieval update/dynamics fixed points. This is exactly my point—fixed-point convergence and energy convergence are not the same thing.
> > > >
> > > > While it’s hard to engineer, it could occur in certain configurations of stored memory patterns. For completeness and rigor in theoretical work, it's important to address this. If a theory doesn't hold under certain scenarios, one should clarify the assumptions or provide further theoretical justifications, as mentioned in [1].
> > > >
> > > > Thank you for the clarification. I have no further issues with Thm2. I suggest the authors to polish the draft according to our discussion and include a remark on the parameter region where it's not vacuous. It will clarify things for general audience.
> > > >
> > > > Thank you! Good luck!

---

### Official Review · Reviewer_wQ3r · 2024-06-25

**Soundness:** 3
**Presentation:** 2
**Contribution:** 2
**Rating:** 4
**Confidence:** 3

**Summary:**

The paper offers a kernel-based approximation of Dense Associative Memory that allows for Hebbian encoding in a space of randomized basis functions. The main advantage when compared with the exact approach is that information concerning all patterns is shared in a single weight tensor, without requiring additional weights for each new pattern.

The authors provide some complexity results and bounds on the Euclidean deviation from the exact retrieved pattern in the case of exponential Dense Associative Networks (i.e. modern Hopfield networks). In a series of numerical experiments, the author show that the approximation breaks down for low values of the inverse temperature and for large number of patterns when compared with the number of basis functions.

**Strengths:**

1) The paper addresses perhaps the greatest limitation of Dense Associative Memories with non-quadratic energy, namely the fact that they cannot compress the patterns into a single shared weight matrix. This has potentially great practical relevance for the implementation of these models in practical applications. Furthermore, it connects the theory of non-quadratic Dense Associative Memories with a very large body of theoretical research on Hebbian learning and classical Hopfield models.

2) The proposed approximate algorithm is theoretically sound as it connects with important known results on kernel machines and feature expansions.

3) The exposition is clear and it is relatively easy to follow even for non-specialists.

4) The numerical analysis of the error profile of the approximation error is thorough and insightful, especially when interpreted in combination with the theoretical results.

**Weaknesses:**

1) The embedding of this paper in the existing literature is limited. Feature expansion methods have been studied for classical Hopfield and generalized models [1,2,3], and there are uncited existing papers connecting Dense Associative Memories and kernel methods [4]. It would also be useful to discuss the recent results connecting exponential Dense Associative Memories (i.e. modern Hopfield networks) with generative diffusion models [5], since as pointed out in the paper the connection provides an alternative way to learn the same energy function in a fixed synaptic structure. It could also be useful to discuss more theoretical results involving random features, for example [6] and [7].
It would be good to have this treatment organized in a proper Related Work section.

2) While the approximation is based on a sound idea, its derivation is very informal. It would be very useful to derive the formula using variational techniques or other known systematic methods for deriving approximations. It could also be interesting to connect it with similar approximations in Gaussian Processes, since kernel feature expansions are commonly used for GPs.

3) I am not convinced about the usefulness of the bound in Eq.12 in non-trivial regimes due to the exponential dependence on the energy.

4) Given the simplicity of the proposed method and its similarities with known approaches such as [TODO], I am not convicted that the contribution is relevant enough for this conference.

References:
[1] Liwanag, Arnold, and Suzanna Becker. "Improving associative memory capacity: one-shot learning in multilayer Hopfield networks." Proceedings of the 19th Annual Conference of the Cognitive Science Society. Vol. 442. 1997.
[2] Barra, Adriano, Matteo Beccaria, and Alberto Fachechi. "A new mechanical approach to handle generalized Hopfield neural networks." Neural Networks 106 (2018): 205-222.
[3] Yilmaz, Ozgur. "Machine Learning Using Cellular Automata Based Feature Expansion and Reservoir Computing." Journal of Cellular Automata 10 (2015).
[4] Uniform Memory Retrieval with Larger Capacity for Modern Hopfield Models
[5] Ambrogioni, Luca. "In search of dispersed memories: Generative diffusion models are associative memory networks." arXiv preprint arXiv:2309.17290 (2023).
[6] Negri, Matteo, et al. "Random Feature Hopfield Networks generalize retrieval to previously unseen examples." Associative Memory {\&} Hopfield Networks in 2023. 2023.
[7] Negri, Matteo, et al. "Storage and learning phase transitions in the random-features hopfield model." Physical Review Letters 131.25 (2023): 257301.

**Questions:**

1) I find intuitively strange that the bound in Eq.13 depends on the absolute value of the energy instead of on a relative energy. Can you provide some intuition for this result?

**Limitations:**

The limitations of the proposed approximate method are adequately discussed.

---

> ### Author Rebuttal · Authors · 2024-08-06
>
> > Improve embedding of this paper into existing literature
>
> Thank you for your suggestions, we will update the paper with the Related Work section and will include the discussion of all the papers that you have suggested. As a side note, the reference [4] actually is cited in our submission, please see Ref [16]. Connection with diffusion, Ref [5] is a beautiful work, which we will be delighted to highlight. As well as the theoretical work on random features Refs [6], [7].
>
> > It would be very useful to derive the formula using variational techniques… It could also be interesting to connect it with similar approximations in Gaussian Processes…
> >
>
> We thank the reviewer for these interesting suggestions and connections.
>
> Our presentation of the approximation was motivated by the simplicity of the derivation that highlights how the energy can be approximated with random features in a cleanly straightforward manner; we were hoping to highlight how easily this distributed representation of memories simplifies the energy function. However, after the derivation, we formally and precisely present how the energy descent can be performed with these random features in Algorithm 1, carefully controlling the peak memory usage and computation. Naively, the peak memory usage would have been $O(YD)$, while our carefully crafted algorithm only requires $O(Y+D)$ peak memory with the same outcome.
>
> However, we do think it is important to derive this approximation via other techniques, potentially highlighting other avenues for improvement.
>
> We thank the reviewer for connecting our work to Gaussian processes. It is true that random features have been used for approximating Gaussian Processes, but the form of usage is targeted to a different application. Gaussian Processes are often used for supervising learning (either directly, or as part of a black-box derivative-free optimization), and the main bottleneck is the need to compute and invert the kernel gram matrix of the training points for any inference. With random features, we can instead perform Bayesian linear learning in the expanded feature space, obviating the kernel gram matrix computation and inversion.
>
> DenseAMs are often utilized for memory retrieval, which is  different than supervised learning (the main application of Gaussian Processes). However, this suggested connection by the reviewer can lead to interesting cross-contribution between the fields of Gaussian Processes and DenseAMs. This is very much in line with our goal in this paper to view DenseAMs through a lens of random features and kernel machines, bringing together two fields.
>
> > I am not convinced about the usefulness of the bound in Eq.12 in non-trivial regimes due to the exponential dependence on the energy.
> >
>
> It is true that the divergence upper bound depends on the term $\exp(\beta E(\mathbf{x}))$. However, note that, for the energy function defined in Eq (11), assuming that all memories and the initial queries are in a ball of diameter $1$, $E(\mathbf{x}) \leq \frac{1}{2} - \frac{\log K}{\beta}$, implies that $\exp(\beta E(\mathbf{x})) \leq \exp(\beta/2) / K$.
>
> An important aspect of our analysis is that the bound is query specific, and depends on the initial energy $E(\mathbf{x})$. As discussed above, this can be upper bounded uniformly, but our bound is more adaptive to the query $\mathbf{x}$.
>
> For example, if the query is initialized near one of the memories, while being sufficiently far from the remaining $(K-1)$ memories, then $\exp (\beta E(\mathbf{x}))$ term can be relatively small.
>
> More precisely, with all memories and queries lying in a ball of diameter $1$, let the query be at a distance $r < 1$ to its closest memories, and as far as possible from the remaining $(K-1)$ memories. In this case, the initial energy $E(\mathbf{x}) \approx -(1/\beta) \log [\exp(-\beta r / 2) + (K-1) \exp(-\beta/2)]$, implying that
>
> $$
> \exp(\beta E(\mathbf{x})) \approx \frac{\exp(\beta r / 2)}{\Big[1 + (K-1)\exp(-\beta(1 - r)/2)\Big]} \leq \exp(\beta r / 2)
> $$
>
> For sufficiently small $r < 1$, the above quantity can be relatively small. If, for example, $r \sim O(\log K)$, then $\exp(\beta E(\mathbf{x})) \sim O(K^{\beta/2})$, while $r \to 0$ gives us $\exp(\beta E(\mathbf{x})) \to O(1)$. This highlights the adaptive query-dependent nature of our analysis.
>
> So it is not directly clear why this is not a “non-trivial regime”. We believe that this form of analysis is novel, and highlights the effect of the different quantities of interest in this formulation (such as the effect of the number of memories, energy descent steps, random features, initial particle energy). We accept that this upper bound might not be the tightest. However, that limitation does not necessarily make the analysis “trivial”.
>
> > Given the simplicity of the proposed method and its similarities with known approaches such as [TODO]…
> >
>
> We would be happy to elaborate on this comment if the reviewer kindly specifies the “[TODO]” approaches. We do believe, however, that our approach is novel and that our contribution is relevant to this conference. We will add a Related Work section to better differentiate it from existing prior work.
>
> > I find intuitively strange that the bound in Eq.13 depends on the absolute value of the energy instead of on a relative energy. Can you provide some intuition for this result?
> >
>
> This result uses the form of the energy, given by Eq (11), which assumes that the zero-energy state is chosen such that the sum of the exponents under the logarithm is equal to 1. Only the difference of the energies appears in Eqs (12,13), but this difference is implicit, given the (arbitrary) choice of the reference point made in Eq (11).

---

> > ### Author Response · Authors · 2024-08-12
> >
> > With the discussion period soon drawing to a close, we wanted to check in to see if our rebuttal has satisfactorily addressed the concerns you raised in your initial review? If you have any further questions or require additional clarifications, we would be more than happy to engage in further discussion.

---

> > > ### Comment · Reviewer_xsUY · 2024-08-13
> > > **raising my score**
> > >
> > > I'm satisfied with the answers and raising my score.

---

### Official Review · Reviewer_B4qp · 2024-07-14

**Soundness:** 2
**Presentation:** 3
**Contribution:** 3
**Rating:** 7
**Confidence:** 2

**Summary:**

This paper studies a method to modify associative memory network weights when introducing new memories. The proposed uses random features and is shown to approximate the energy function of the conventional ones.

**Strengths:**

1. The proposed method and results are novel to me, understanding associative memories (AMs) through random feature seems to be a great connection between AMs and random feature transformers.
2. The paper is well written and easy to follow.

**Weaknesses:**

1. Why using hamming distance when calculating retrieval error when the theoretical results use L2?
2. To my understanding, the proposed method is an approximation to DAM. Thus the paper should at least compare the proposed method to DAM in thr experimental section.
3. The field of associative memories, Hopfield networks are getting increased attention recently, I think a related work section would benefit this paper a lot.

**Questions:**

Please refer to weaknesses.

**Limitations:**

This paper addresses its limitations in the appendix.

---

> ### Author Rebuttal · Authors · 2024-08-06
>
> > Why using hamming distance when calculating retrieval error when the theoretical results use L2?
> >
>
> We consider Hamming error since we are storing and retrieving binary memories, e.g. in Fig.3. Our theoretical results operate in the more general continuous $\mathbb{R}^d$ space, and thus bounds the L2 error. However, please note that, with binary vectors, Hamming error and L2 error are related by the square root operation. For this reason, our theoretical bound (proven for L2 error) is equal to the square root of the numerical evaluations with Hamming distance.
>
> > To my understanding, the proposed method is an approximation to DAM. Thus the paper should at least compare the proposed method to DAM in the experimental section.
> >
>
> You are correct -- our proposed "distributed representation" for DenseAMs, which we call DrDAM, are approximations to the original DAM (what we refer to as the "memory representation" or MrDAM in our paper). The experiments in our paper are explicitly designed to compare against the original DAM a.k.a. MrDAM. E.g., Figs 2 and 3A plot DrDAM's approximation error when compared to the original DAM. Fig 3B shows how well, qualitatively, DrDAM approximates the original DAM (what we call the "Ground Truth" in that figure).
>
> > The field of associative memories, Hopfield networks are getting increased attention recently, I think a related work section would benefit this paper a lot.
> >
>
> Thank you for your suggestion. This sentiment was echoed by other reviewers and we will update the paper with a pertinent related works section.

---

> > ### Comment · Reviewer_B4qp · 2024-08-12
> >
> > Thanks for the response.
> > I think the authors have addressed all my concerns, I will raise my score accordingly.

---

### Official Review · Reviewer_xsUY · 2024-07-15

**Soundness:** 3
**Presentation:** 3
**Contribution:** 2
**Rating:** 6
**Confidence:** 3

**Summary:**

The paper proposes to interpret the energy function of Dense Associative Memory as relying on a kernel that can be approximated by kernel-specific random feature maps.
The approximation allows to condensate the stored patterns into a tensor whose size is independent of the number of patterns stored, similarly to what can be done in classical and polynomial Hopfield networks.
New patterns can then be incorporated without changing the dimensionality of the parameter space.
They provide bounds for the approximation error of the iterates in the gradient descent procedure that uses their energy approximation.

**Strengths:**

- I appreciated the idea of trying to use the kernel approximation in order to overcome the computational limits of DAMs.
- The paper is well written and the idea is clearly explained.

**Weaknesses:**

- The main limitation of the work in my opinion is that one has to go to very  large values of the feature size Y in order to have a good approximation.  In fact, looking at the error bound in eq. (12), it seems that Y has to be taken of the order of O(K^2*D) in order to have a good approximation, which is much larger than the size of the memory matrix (K*D). This not taking into account the exp(beta*E) term in the bound, which could penalize high-energy configurations a lot.

- Numerical Experiments are limited and not fully convincing. For instance, in Fig. 1 the 4x12288=49152 memory matrix has to be replaced by a several times larger Y-sized vector in order to obtain a good approximation.

**Questions:**

- Can the author show a plot similar to Fig. 2 but now showing the theoretichal prediction from eq. 12 (say for L=1 and alpha=1) compared to the actual error?
Probably this information could just be plotted in Fig. 2B.

- It would be nice to make a plot similar to Fig. 2 but now as a function of K.

- In order to help the reader and improve consistency, p in Algorithm 1 and at the beginning of Section 3 could denote the same thing.

- Assuming that the random feature version is considered as DAM per se instead of an approximation to another DAM, could the author comment on its capacity in presence of random memories? It is linear in Y?

- line 215. Usually, the EDP model also contains an x^2 term outside the logarithm.

**Limitations:**

The authors could discuss more the limitations of their work. In particular, the fact that the approximation requires a very large feature size Y in order to be accurate. This could limit the practical use of the method.

---

> ### Author Rebuttal · Authors · 2024-08-06
>
> We are happy to hear that the main message of the paper got across well, and thank the reviewer for their insightful comments and feedback. Below we answer individual questions raised.
>
> > … looking at the error bound in eq. (12), it seems that $Y$ has to be taken of the order of $O(K^2 D)$…
> >
>
> It is true that if we ignore all the terms on the right hand side of equation (12) except for the $K \sqrt{D/Y}$ term, it appears that $Y \sim O(K^2 D)$. However, it is important to note that the remainder terms in the upper bound on the right hand side of (12) also include the step-size $\alpha$ and $K$ itself. Ignoring the effect of $\alpha$ and $K$ in the remainder terms can give a wrong impression regarding the bound. For that very reason, we explicitly study a special case in Corollary 1, in which the update rate $\alpha$ is scaled in a $K, L$, and $\beta$ dependent manner. An important aspect of our analysis is that the bound is query specific, and depends on the initial energy $E(\mathbf{x})$. For example, if the query is initialized near one of the memories, while being sufficiently far from the remaining $(K-1)$ memories, then $\exp (\beta E(\mathbf{x}))$ term can be relatively small. More precisely, with all memories and queries lying in a ball of diameter $1$, let the query be at a distance $r < 1$ to its closest memory, and as far as possible from the remaining $(K-1)$ memories. In this case, the initial energy $E(\mathbf{x}) \approx -(1/\beta) \log [\exp(-\beta r / 2) + (K-1) \exp(-\beta/2)]$, implying that
>
> $$
>  \exp(\beta E(\mathbf{x})) \approx \frac{\exp(\beta r / 2)}{\Big[1 + (K-1)\exp(-\beta(1 - r)/2)\Big]} \leq \exp(\beta r / 2)
> $$
>
> Thus, the exponential factor in our bound is not a problem. Overall, the divergence is bounded by $O(\sqrt{D / Y})$, at constant $\beta$, implying the need for $Y \sim O(D / \epsilon^2)$ for at most $\epsilon$ divergence. Please note, that this estimate is independent of $K$, which is the essence of our proposal.
>
> > Numerical Experiments are limited… in Fig. 1 the 4x12288=49152 memory matrix has to be replaced by a several times larger Y-sized vector…
> >
>
> The configuration shown in Fig.1 was chosen to illustrate the main idea of the method. $Y$ does not have to be larger than the number of parameters required to describe the memory vectors. To explicitly demonstrate this point we have repeated the experiments shown in Fig.1 with 20 memories; please see Fig (a) in the 1 page PDF rebuttal. Now we have 20x64x64x3=246k parameters in the memory vectors encoded in the 200k-dimensional vector $T$. The method works well and all the conclusions presented in the paper remain valid.
>
> > Can the author show a plot similar to Fig. 2 but now showing the theoretical prediction from eq. 12…?
> >
>
> We thank the reviewer for this great suggestion, which would highlight the tightness of our proposed upper bound. We have done it in Fig (c) of the 1 page PDF rebuttal. It is important to note that the upper bound also involves a quantity (precisely the quantity $C_1$ defined in assumption A2 in Theorem 2) which bounds the approximation introduced by the random feature maps (such as Rahimi and Recht (2007)), and this does not have an easily computable analytical form. For this reason, it is challenging to precisely compute this bound. However, it is possible to check the predicted scaling relationships. Specifically, the theoretically predicted $Y^{-1/2}$ dependence in the upper bound for a fixed $D$ appears in most of the results, please see Fig (c) in the 1 page PDF. This shows that the upper bound  in equation (12) from Theorem 2 fairly characterizes the dependence on $Y$. Fig (c) from the PDF will be used to update Fig 2 (right side) in the current submission.
>
> > It would be nice to make a plot similar to Fig. 2 but now as a function of K.
> >
>
> Thanks for the suggestion. We have generated this plot per your request and included it in Fig (b) of the 1 page PDF. The results are consistent with the theory developed in our paper.
>
> > … $\mathbf{p}$ in Algorithm 1 and at the beginning of Section 3 could denote the same thing.
> >
>
> The reviewer is correct that we overload the variable $\mathbf{p}$ both as $\mathbf{p} = \mathbf{g}(\mathbf{x})$ and $\mathbf{p} = RF(\tau, \mathbf{g}(\mathbf{x}))$. We will fix the notation, and use a different term to denote $\mathbf{g}(\mathbf{x})$.
>
> > line 215. Usually, the EDP model also contains an $x^2$ term…
> >
>
> The reviewer is correct, the complete EDP energy is
>
> $$
> E(\mathbf{x}) = -\frac{1}{\beta} \log \sum_\mu \exp(\beta \langle \boldsymbol{\xi}\mu , \mathbf{x} \rangle) + \frac{1}{2} \| \mathbf{x} \|^2.
> $$
>
> We were referring to the part of the energy which is relevant for the argument in line 215. We will write the complete energy. Our claim of validity of the proof technique of Theorem 2 still applies, since we just need to approximate the $\exp(\beta\langle \boldsymbol{\xi}_\mu, \mathbf{x} \rangle)$ term with the random features.
>
> > Assuming that the random feature version is considered as DAM… could the author comment on its capacity in presence of random memories?
> >
>
> This is an interesting question, which may be investigated in the future. However, it remains beyond the scope of our current project. Our goal here is to answer the following question: how well the random feature description can represent the dynamics of DAM?
>
> **General Comment**
>
> Once again, we greatly appreciate your insightful comments and suggestions. We would be grateful if you could consider increasing the score for our work towards an accept.

---

> > ### Author Response · Authors · 2024-08-12
> >
> > With the discussion period soon drawing to a close, we wanted to check in to see if our rebuttal has satisfactorily addressed the concerns you raised in your initial review? If you have any further questions or require additional clarifications, we would be more than happy to engage in further discussion.

---

### Author Rebuttal · Authors · 2024-08-06

We sincerely thank all reviewers for their thoughtful feedback. We are honored that our work has been recognized as addressing “the greatest limitation of Dense Associative Memories with non-quadratic energy” [wQ3r] and as making a “significant and solid step forward for the community” [9NBh]. We are also grateful that reviewers have acknowledged the clarity and accessibility of our paper [xsUY, B4qp, wQ3r].

Reviewers have identified several areas in our presentation that need improvement:

1. **A more complete list of related works** [wQ3r, 9NBh, B4qp]**.**  We appreciate the reviewers’ suggestions for additional relevant works. We have reviewed the related work suggested by reviewers and find them all relevant, and will enhance the “Related Works” section accordingly to more thoroughly embed our work into the context of existing studies.
2. **Improve the motivation for the method** [xsUY, 9NBh]. We understand that reviewers are looking for compelling empirical reasons to use the distributed representation for DenseAMs over the traditional memory representation, particularly in terms of parameter efficiency or time complexity. We hope that our additional experiments show that information compression is indeed possible using DrDAM; however, we want to emphasize that the main message of our paper is to show and characterize how we can uncouple the actual memory patterns from the definitions of (1) the energy and (2) the full memory-retrieval dynamics of DenseAMs.
3. **Additional experiments** [xsUY, 9NBh, B4qp] We have run additional experiments and included them in the attached 1-page PDF, further validating the theory and practicality of our method:
    - Fig (a): We designed Fig 1 of the original submission as an illustration of the method, but Reviewer xsUY astutely pointed out that in this experiment our distributed representation method (DrDAM) used more parameters than the memory representation for DenseAMs (MrDAM). We have thus updated that experiment to store and retireve 20 images from TinyImagenet into both DrDAM and MrDAM configured at the same $\beta$, showing that DrDAM can successfully compress information in the stored patterns. Full details are provided in the caption and app. C of the orig. submission
    - Fig (b): In Fig 2 of the original submission we analyzed the error between DrDAM and MrDAM as a function of $Y$ (the size of the distributed tensor $\mathbf{T}$), $D$ (the size of queries and stored patterns), and $\beta$ (the inverse temperature). We additionally performed this error analysis as a function of how many patterns $K$ are stored in the memory. See more details in the caption.
    - Fig (c): Here we verify the tightness of the error bound (a function of $\frac{1}{\sqrt{Y}}$ ) in Thm. 2 on top of the empirical curves of Fig 2. We will update the right part of Fig 2 of the original submission with this version upon acceptance.

We have addressed each reviewer’s individual questions and concerns below. We respectfully ask each reviewer to consider increasing their score if concerns have been satisfactorily addressed.

---

### Author Response · Authors · 2024-08-13

Dear Reviewers, Dear Area Chair,

Once again, thanks a lot for taking the time to read our work and to communicate to us your feedback. On reading the reviews, we are very happy to see many enthusiastic statements, e.g., "The paper addresses perhaps the greatest limitation of Dense Associative Memories with non-quadratic energy" from reviewer wQ3r. Such statements mean a lot to us, and we are thankful for these encouraging comments.

There is still one borderline reject score (4) from reviewer wQ3r, and we have not heard from them any response to our rebuttal. Since there is less than 24 hours left for us to be able to communicate with reviewers, we kindly ask reviewer wQ3r to consider increasing the numerical score for our submission.

We are thankful to reviewers B4qp and xsUY for raising their scores to 6 and to reviewer 9NBh for raising their score to 7! With the decision-making process soon underway, if B4qp or xsUY believe our paper is worthy of acceptance, we kindly ask considering a further increase of the scores to a "full accept" (7). We remain available to resolve any outstanding issues that would bump our paper from a borderline accept to an accept in the remaining time dedicated for discussion.

Sincerely,
Authors

---

### Decision · Program_Chairs · 2024-09-25

**Decision:**

Accept (poster)

**Comment:**

The present study proposes a kernel approximation of the dense associative memory. Connecting the random feature approximation and the dense associative memory is novel. After comprehensive author-reviewer discussions, reviewers have carefully checked the mathematical derivations in the paper and the authors' replies have addressed most of the concerns. Therefore I recommend accepting the present paper. Meanwhile, reviewers pointed out the original manuscript hasn't systematically compared with related work and fully discussed its limitations. Please revise the manuscript based on the reviewers' comments.